

# FLAME 1.0 : a novel approach for modelling burned area in the Brazilian biomes using the Maximum Entropy concept

Maria Lucia Ferreira Barbosa[1,2,*], Douglas I Kelley[3], Chantelle A Burton[4], Igor J M Ferreira[1,5], Renata Moura da Veiga[1], Anna Bradley[4], Paulo Guilherme Molin[2], Liana O Anderson[6]

[1] National Institute for Space Research - INPE, Avenida dos Astronautas, 1758. Jd. Granja - São José dos Campos - São Paulo, 12227-010, Brazil

[2] Federal University of São Carlos, Rodovia Lauri Simões de Barros, km 12 - SP-189 - Aracaçu, Buri - São Paulo, 18290-000, Brazil

[3] UK Centre for Ecology and Hydrology, Wallingford. OX10 8BB UK

[4] Met Office Hadley Centre, Fitzroy Road, Exeter. EX1 3PB UK

[5] Faculty of Environment, Science and Economy, University of Exeter, Exeter, UK

[6] National Centre for Monitoring and Early Warning of Natural Disasters - Cemaden, Estrada Doutor Altino Bondensan, 500 - Distrito de Eugênio de Melo, São José dos Campos - São Paulo, Brazil

* Correspondence to: Maria Lucia Ferreira Barbosa (malucsp@gmail.com)

**Abstract**

As fire seasons in Brazil lengthen and intensify, the need to enhance fire simulations and comprehend fire drivers becomes crucial. Yet determining what drivers burning in different Brazilian biomes is a major challenge, with the highly uncertain relationship between drivers and fire. Finding ways to acknowledge and quantify that uncertainty is critical in ascertaining the causes of Brazil's changing fire regimes. We propose FLAME (Fire Landscape Analysis using Maximum Entropy), a new fire model that integrates Bayesian inference with the Maximum Entropy (MaxEnt) concept, enabling probabilistic reasoning and uncertainty quantification. FLAME utilizes bioclimatic, land cover and human driving variables to model fires. We apply FLAME to Brazilian biomes, evaluating its performance against observed data for three categories of fires: all fires (ALL), fires reaching natural vegetation (NAT), and fires in non-natural vegetation (NON). We assessed burned area responses to variable groups. The model showed adequate performance for all biomes and fire categories. Maximum temperature and precipitation together are important factors influencing burned area in all biomes. The number of roads and amount of forest boundaries (edge densities), and forest, pasture and soil carbon showed higher uncertainties among the responses. The potential response of these variables displayed similar spatial likelihood of the observations given the model, between the ALL, NAT and NON categories. Overall, the uncertainties were larger for the NON-category,





particularly for Pampas and Pantanal. Customizing variable selection and fire categories based
on biome characteristics could contribute to a more biome-focused and contextually relevant
analysis. Moreover, prioritizing regional-scale analysis is essential for decision-makers and fire
management strategies. FLAME is easily adaptable to be used in various locations and periods,
serving as a valuable tool for more informed and effective fire prevention measures.
Keywords: Burned Area. Brazilian biomes. Maximum Entropy. Bayesian Inference. Climate.
Fragmentation. Land Use.

**1 INTRODUCTION**

The complexity of the interactions and feedbacks between fire, climate, people, and other earth
system components makes it challenging to be highly confident about what drives fires in
specific locations. Various methods assess the drivers of historical fire events. Some studies
correlate individual drivers with burned area but overlook the interaction of multiple factors
(ANDELA et al., 2017; BARBOSA et al., 2019). Fire Danger Indices capture simultaneous
drivers to gauge fire risk. However, they overlook human-driven ignition causes
(ZACHARAKIS; TSIHRINTZIS, 2023) and typically fail to capture the impact of fuel
availability on burning (KELLEY; HARRISON, 2014). Fire-enabled Land Surface Models
account for these drivers, simulating observable fire regime measures. However, they often
lack accuracy for year-to-year fire patterns and required accuracy to determine fire drivers
(FORKEL et al. 2019) and the causes of individual fire seasons (HANTSON et al., 2020).
Quantifying uncertainty is critical for assigning fire drivers because it allows for a more
accurate assessment of the confidence in our predictions and helps identify the most influential
factors under varying conditions. In this sense, research applying the Maximum Entropy
framework combined with Bayesian Inference can address these gaps.

The Principle of Maximum Entropy (MaxEnt) states that when trying to estimate the
probability of an event and the information is limited, you should opt for the distribution that
preserves the greatest amount of uncertainty (i.e., maximizes entropy) while still adhering to
your given constraints (PENFIELD, 2003). These constraints reflect prior knowledge about the
probability distribution of a phenomenon of interest (i.e., burned area) based on its relationship
with independent variables. This approach ensures you do not introduce extra assumptions or
biases into your calculations. MaxEnt has its roots in statistical mechanics (JAYNES, 1957).



However, the use of its concept in a species distribution model (PHILLIPS et al., 2006)
popularized the approach in several other study areas, including ecology, geophysics, and fires
( JIN et al., 2020;  LI et al., 2019; FONSECA et al., 2017).  Incorporating Bayesian Inference
alongside the MaxEnt framework further enhances this approach. Bayesian techniques
integrate prior knowledge and observed data to continuously refine the estimation of
uncertainty in the influence of drivers on fire, thereby improving the confidence in a
relationship we find. By leveraging both MaxEnt and Bayesian Inference, we can develop more
robust models that account for the complex and dynamic nature of fire regimes.

The MaxEnt species distribution model estimates the probability of target presence for given
local conditions (PHILLIPS et al., 2006). Unlike many traditional models, MaxEnt makes
minimal assumptions about the relationships between variables, making it more flexible and
adaptable to complex ecological interactions. Rather than estimating a single value, MaxEnt
models a full probability distribution (ELITH et al., 2011), providing a comprehensive view of
potential outcomes. This probabilistic nature enables the incorporation of prior information
into the modeling process, enhancing its accuracy. Additionally, MaxEnt enables the
quantification of uncertainties (CHEN et al., 2019), providing valuable insights into the
reliability and confidence of model predictions.

Recognizing that fires can be treated as a species due to their strong dependence on
environmental factors, utilizing the MaxEnt species model has yielded valuable insights into
the field (FERREIRA et al., 2023; FONSECA et al., 2019). However, the MaxEnt model relies
on presence-only or presence/absence data, which means it primarily considers locations where
the target (in this case, fires) has occurred. This limits fire research using MaxEnt as it does not
allow continuous data, such as burned area fraction over a larger region. Moreover, the
constraints and structure of the underlying model are fundamentally related to species
distributions (PHILLIPS et al., 2006) rather than fires, which may not capture the nuances of
fire behavior.

The simulation of fires in heterogeneous territories such as Brazil is incredibly challenging.
Wildfires have become a pressing concern in the country, causing significant socioeconomic
and environmental losses (CAMPANHARO et al., 2019; BARBOSA et al., 2022; WU et al.,
2023). Since 1980, more than 1,857,025 km² of Brazil's terrain has been negatively impacted
by fires (MAPBIOMAS, 2023), reflecting a need for effective and adaptive fire management





strategies. Nonetheless, quantifying the influence of these drivers can be difficult - many
interactions between fire and its drivers are non-linear, and drivers heavily interact with each
other, making confidently identifying drivers of fire regimes in such diverse landscapes tricky
from observations alone (KRAWCHUK and MORITZ 2014). While traditional fire models
provide useful broadscale information on fire, land, and climate interactions, they do not
quantify the uncertainty in these relationships and rely on other studies to infer relationships
between drivers and burning (HANTSON et al., 2016).

Improving fire simulations and understanding the underlying drivers of fires in Brazil is
essential to address the challenges associated with preventing fires, firefighting, and managing
their aftermath. Here, we present and evaluate a novel fire model, FLAME (Fire Landscape
Analysis using Maximum Entropy), based on a Bayesian inference implementation of the
MaxEnt concept. This combination allows us to incorporate uncertainty and probabilistic
reasoning into fire modeling. In this sense, the model aims to precisely measure uncertainties
of the simulations. The model optimizes key driving variables relationship with fires. Here we
apply FLAME to the biomes in Brazil, and assess the performance against observations.

**2 METHODS**

**2.1 Datasets and preprocessing**

We used the MCD64A1 burned area product from MODIS collection 6 as our target variable
(GIGLIO et al., 2018). This data was regridded from 500m to 0.5° spatial resolution.The burned
area data was used in its totality (ALL) and divided into two other categories based on the
LULC data from the Mapbiomas project (https://brasil.mapbiomas.org/en/): fires reaching
natural vegetation (NAT) and fires reaching non-natural vegetation (NON) (Fig. 1).

We computed all burned areas within forests, grasslands, and savannas for the NAT and the
NON within pasture, cropland, and forest plantation, aggregated with croplands. The
categorization of fires aims to assess whether there are distinct drivers for NAT and NON and
to exemplify the potentialities of the model for assessing more than one fire category across
different vegetation types. Amazonia and Atlantic Forest are fire-sensitive biomes (Fig. 1) that
are highly susceptible to damage or destruction by fire. Cerrado, Pampa and Pantanal have





evolved to depend on fire as part of their life cycle and are considered fire-dependent biomes.
Finally, Caatinga is a fire-independent biome that is generally not significantly affected by fire
or does not require fire as part of its vegetation dynamics. This categorization follows Hardesty
et al. (2005), based on the predominant vegetation type that defines the biome. However, all
biomes contain vegetation types with different sensitivities to fire. We adopt a broad approach
to encompass the various biomes in Brazil; however, any type of categorization is permissible,
and further studies could focus on even finer stratification, e.g. fires reaching fire-sensitive
vegetation and fire-dependent vegetation within each biome.

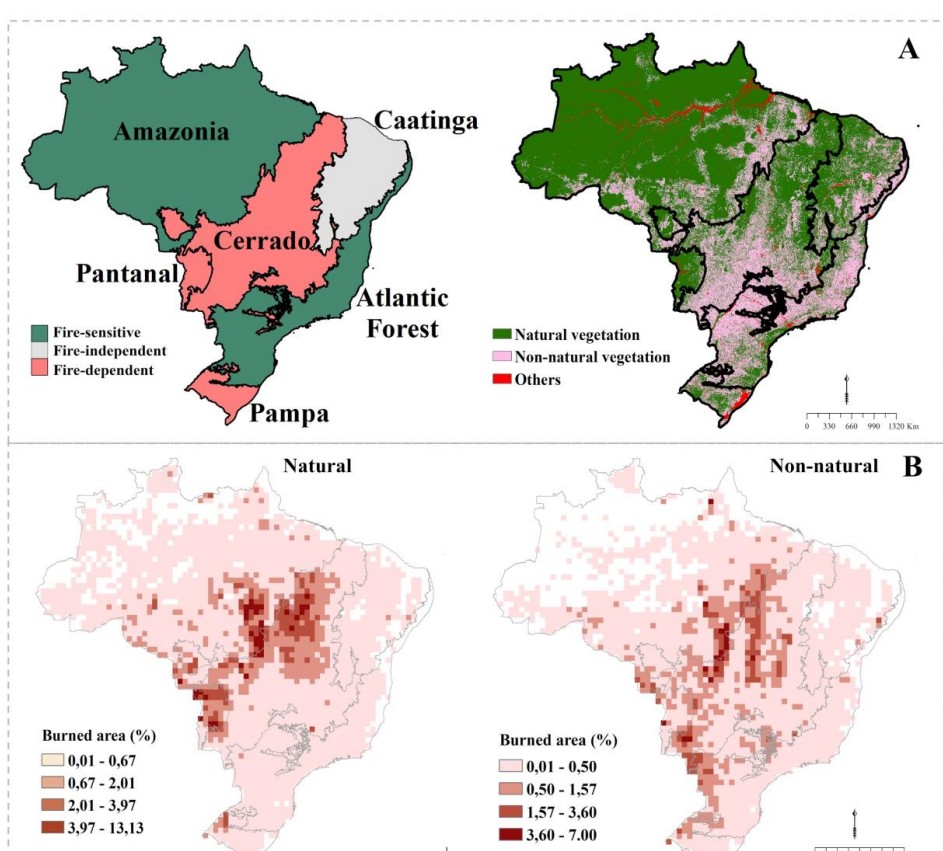


**Figure 1: (A) Brazilian biomes classified as Fire-sensitive, Fire-independent and Fire-**
**dependent on the left (HARDESTY et al. 2005) and Natural vegetation (Forests,**
**Grasslands and Savannas) and Non-natural vegetation (Pasture, Cropland and Forest**
**Plantations) in 2019 in Brazil on the right.  (B) NAT's mean burned area percentage per**



**pixel is on the left and NON is on the right. The maps show the mean for August,**
**September and October from 2002 to 2019.**

The target and independent variables were extracted for August, September, and October, from
2002 to 2019, representing the general peak of the fire season in Brazil. This time frame is the
most extended overlapping period between the datasets which we further divided into a training
phase from 2002 to 2009 and a validation phase from 2010 to 2019. The independent variables
were divided into five groups (climate, anthropogenic and natural ignition, fuel, LULC and
forest metrics) and are described in Table 1. We acquired climate variables from the first
component of the third simulation round of the Inter-Sectoral Impact Model Intercomparison
Project (ISIMIP3a, https://www.isimip.org/). ISIMIP is a collaborative effort to compare and
evaluate the outputs of various climate and impact models (FRIELER et al., 2023). This data
represents the historical simulations using climate-forcings from GSWP3-W5E5, available
from 1901 to 2019 at a 0.5° spatial resolution.

We obtained soil, vegetation carbon and soil moisture from the Joint UK Land Environment
Simulator Earth System impacts model at version 5.5 (JULES-ES; MATHISON et al., 2023)
and driven by ISIMIP3a GSWP3-W5E5 as per Frieler et al. (2023), which is freely available
at https://www.isimip.org/impactmodels/details/292/. JULES-ES has previously been used as
input for Bayesian-based fire models (e.g. UNEP et al., 2022). JULES dynamically models
vegetation, carbon fluxes and stores in response to meteorology, hydrology, nitrogen
availability, and land use change. JULES-ES has been extensively evaluated against snapshots
and site-based measurements of vegetation cover and carbon (MATHISON et al., 2023;
WILTSHIRE et al. 2021; BURTON et al.,2019; BURTON et al. 2022). As per UNEP et al.
(2022), vegetation responses to JULES-ES's internal fire model were turned off so as not to
double-count the effects of burning. The maps, therefore, represent environmental carbon
potential and are applicable to FLAME as the model only assumes that variable ranges are
correctly ranked – i.e. areas of low/high carbon content correspond with real-world areas of
low/high carbon and not that the absolute magnitude is correct.

Regarding ignition variables, Population Density data was also obtained from the ISIMIP3a
protocol and based on data from the History Database of the Global Environment (HYDE) v3.3
(VOLKHOLZ et al., 2022). Lightning was prescribed as a monthly climatology from LIS/OTD
data (CECIL, 2006). The LIS/OTD Climatology datasets comprise gridded climatologies that



document the lightning flash rates detected by the Optical Transient Detector (OTD) and the
Lightning Imaging Sensor (LIS) aboard the Tropical Rainfall Measuring Mission (TRMM).
We collected road density data from the Global Roads Inventory Project (GRIP) (MEIJER et
al. 2018), using total density in $m/km^2$, which we regridded to the 0.5-degree grid used by the
rest of the data using linear interpolation in the Iris Python package (MET OFFICE, 2023).

We used the collection 7 LULC data from the MapBiomas project, which produces annual
LULC mapping for the Brazilian territory. They were regridded from 30 m to 0.5∘ to match
the coarser resolution and interpolated from an annual to a monthly time step.

The forest metrics variables were calculated into the 0.5º grid based on the forest data from the
Mapbiomas at 30m resolution using the package 'landscapemetrics' available in R
(HESSELBARTH et al., 2023). The metrics were number of patches (NP) and edge density
(ED):

$$NP = n_i \tag{1}$$

where $n_i$ is the number of patches belonging to class *i*. NP is an 'Aggregation metric' and
describes the fragmentation of a class, in this case, forest formations.

$$ED = \frac{\sum e_i}{A} \tag{2}$$

where $e_i$ is the total edge length in meters, and A is the total landscape area in square meters.
It quantifies edge density by summing up all edges within class *i* in relation to the overall
landscape area. This metric provides insights into the landscape's configuration. We
incorporated these metrics to integrate fragmentation variables - studies suggest that these are
linked to fire occurrence in Amazonia and Cerrado (SILVA JUNIOR et al., 2022; ROSAN et
al., 2022) but remain unexplored in the other biomes.






| Group | Variable | Abbreviation | Source |
|---|---|---|---|
| CLIMATE | Maximum Temperature (ºC) | tmax | ISIMIP3a FRIELER et al. (2023) |
| | Precipitation (m/sec) | ppt | |
| | Vapor pressure deficit (Pa) | vpd | |
| | Relative Humidity (fraction) | rh | |
| | Consecutive number of dry days (days) | dry_days | |
| | Soil Moisture (fraction) | soilM | JULES-ES |
| IGNITION | Lightning (flashes/km/day) | lightn | ISIMIP3a FRIELER et al. (2023) |
| | Population density (people/1000 $km^2$) | pop | |
| | Road density (m/$m^2$) | road | GRIP global (MEIJER et al., 2018) |
| FUEL | Vegetation carbon (kg/m2) | cveg | JULES-ES |
| | Soil carbon (kg/m2) | csoil | JULES-ES |
| LULC | Forest (%) | forest | MAPBIOMAS, 2022 |
| | Grassland (%) | grass | |
| | Savanna (%) | sav | |
| | Cropland (%) | crop | |



| FOREST METRICS | Pasture (%) | pas | |
|---|---|---|---|
| | Number of patches | np | Calculated from MAPBIOMAS, 2022 |
| | Edge density (m/m²) | ed | |

215                 Table 1. Initial list of explanatory variables.

**2.2 Variables selection**

In constructing our predictive model, we considered the interrelationships among different
variables to ensure a robust and coherent analysis. The selection of variables was guided by
their correlation, aiming for a set of features that provided information without redundancy.
For this, we calculated the Spearman correlation coefficient (SPEARMAN, 1961) presented in
Fig. 4.2. We chose Spearman rank over other correlation metrics as our model has a non-linear
relationship between drivers and fires (Section 2.3), making it a better assessment than
parametric comparisons. We identified variables that exhibited strong relationships by
examining the correlation matrix, which we removed from the final model. We used a threshold
higher than 0.6 from Spearman's coefficient for this. The selection was also based on previous
knowledge about the variables relationship with burned area. For example, we did not include
lightning even though it presented low correlation with other variables. Fires caused by
lightning are uncommon and usually occur during the wet season (MENEZES et al., 2022)
which is out of scope of our analysis.






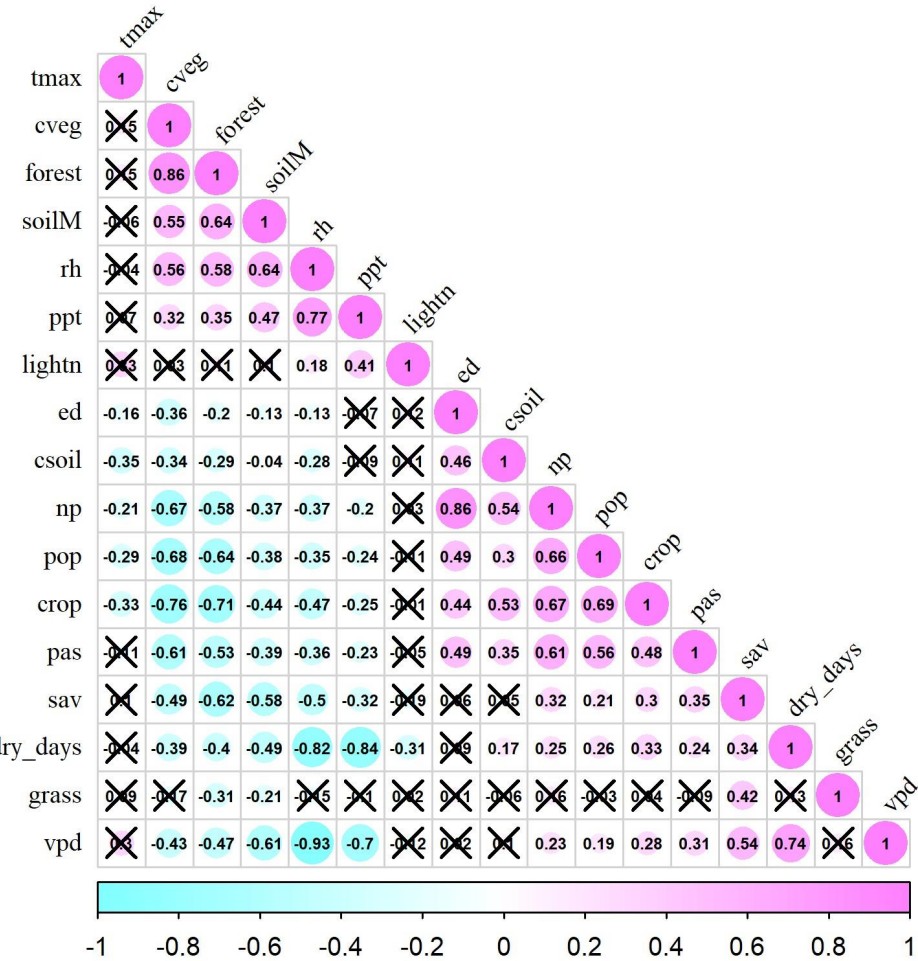

**Figure 2: Spearman correlation of the explanatory variables. Crossed values indicate no correlation, values near 1 [magenta] indicate a strong positive correlation and near -1 [cyan] a strong negative correlation.**

We adopted a more streamlined approach by opting for a shorter list of variables and by grouping them in the variables analysis to capture their compound effect. Initially, we selected 7 variables as input for the final model (Fig. 3) from the 18 initial variables. These variables were chosen based on their correlation, ensuring that at least one variable from each group was selected (Climate, Fuel, LULC, Ignition and Forest Metrics). Next, we divided the variables into three groups. Group 1 is composed of climate variables Maximum Temperature and Precipitation; Group 2 includes the variables Edge Density and Road Density which are related



with landscape fragmentation; and Group 3 encompasses Forest cover, Pasture cover and
Carbon in dead vegetation which are associated with fuel availability.

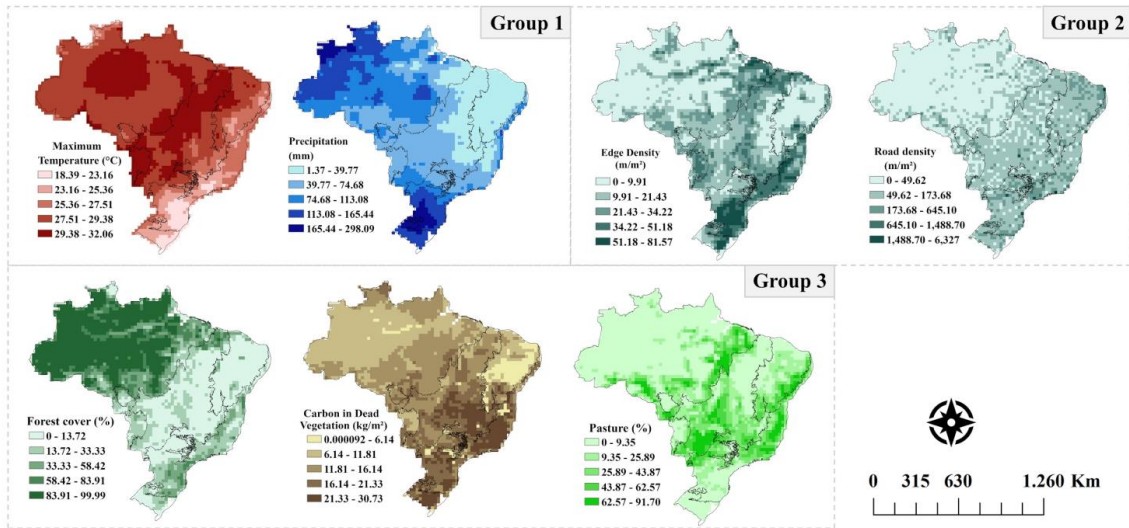

**Figure 3: Mean of the selected explanatory variables for August, September and October**
**from 2002 to 2019.**

**2.3 Relationship curves**

The constraints or priors of the model were added as parameters of different functions, which
we refer to as relationship curves. We included the linear and power functions (Fig. 4)
according to known relationships between fires and environmental variables. This means that
some environmental variables, when presenting higher values, are likely to increase fires. In
comparison, others have an inverse relationship where lower values of the variable coincide
with an increase in burned area. We expect our selected variables to have the following
relationship with fires:
1. Maximum Temperature, Carbon in dead vegetation and Pasture are expected to increase

261         burning with the increase of the variable (CANO-CRESPO et al., 2015; DOS SANTOS

262         et al., 2021; LIBONATI et al., 2022);

2. Precipitation and Forest, which we expect to increase burning with the decrease of the

264         variable (ARAGÃO et al., 2008; BARBOSA et al., 2022);

3. Edge Density and Roads are expected to have more uncertain response across the

266         biomes. High density of edges can lead to more fires into forest ecosystems





(ARMENTERAS et al., 2013; SILVA-JUNIOR et al., 2022) but fragmentation can also
reduce fires by impeding fire spread (DRISCOLL et al., 2021). Regarding Road
Density, while more fires are expected surrounding roads (ARMENTERAS et al.,
2017), less fires are expected with increased density due to urbanization.

The model then estimates the contribution of each curve to the final model. Even though it is
possible to include more relationship curves, we decided to keep it at a minimum to avoid
making too many assumptions and unstable results due to computational efficiency.

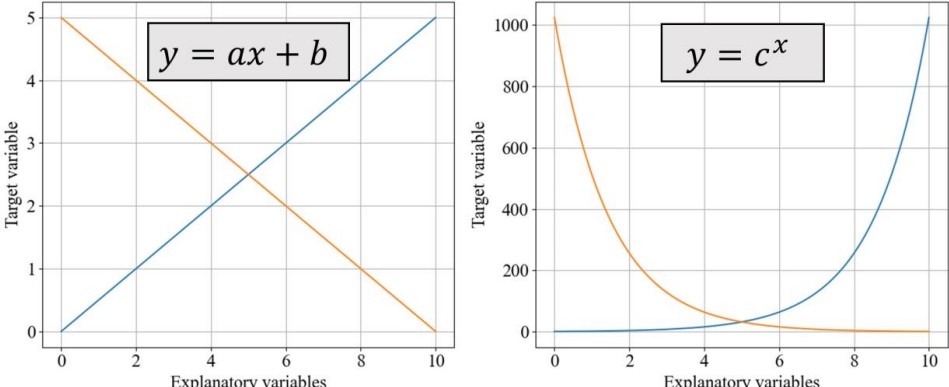


**277 Figure 4: Graphical representation of the relationship functions implemented in the**
**278 model. The one on the left is a linear function and on the right is a power function.**

**2.4 Model optimization**

The model was optimized for each Brazilian biome separately using the MCD64A1 product
from 2002 to 2009. This process used the PyMC5 Python package (ABRIL-PLA et al., 2023),
employing 5 chains each over 1000 iterations using the No-U-Turns Hamilton Monte Carlo
sampler (HOFFMAN and GELMAN 2014) while utilizing 20% of the data or a minimum of
6000 grid cells. While the runs were conducted individually for each Biome, the results were
aggregated to facilitate visualization. The code used to develop this model is available on
GitHub repository (https://github.com/malu-barbosa/FLAME).

In Bayesian inference, we update our beliefs or knowledge about a system or event by
incorporating new evidence or data (LAPLACE, 1820; GELMAN et al., 2013). It allows us to





quantify and update our uncertainty using probability distributions. By maximizing entropy,
we aim to achieve the most unbiased, information-rich distribution that satisfies this prior
knowledge. In this sense, the likelihood (or posterior probability) of the values of the set of
parameters, β, given a series of observations $Obs_i$ and explanatory variables ($X_{iv}$, from section
2.2) is proportional (∝) to the prior probability distribution of $P(\beta)$ multiplied by the
probability of the observations given the parameters tested.

$$P(\beta \mid \{Obs_i\}, \{X_{iv}\}) \propto P(\beta) \times \Pi_i P(Obs_i \mid \{X_{iv}\}, \beta) \qquad (3)$$

Where $\{Obs_i\}$ is a set of our target observations, and i is the individual data point and $\{X_{i\,v}\}$ is
the set of explanatory variables, v, for data point i. The pi notation ($\Pi$) indicates repeated
multiplication. Maximum Entropy in species distribution modeling assumes that individual
observations $(Obs_i)$ are either 1 when there is a fire or 0 when there is not, and that:
$$P(1 \mid \{X_v\}, \beta) = f(\{X_v\}, \beta) \text{ and } P(0 \mid \{X_v\}, \beta) = 1 - f(\{X_v\}, \beta) \qquad (4)$$
Where $P(1 \mid X, \beta)$ is the probability of a fire to occur, $P(0 \mid X, \beta)$ is the probability of no fire.
The term $f(X, \beta)$ is defined below:
$$f(\{X_v\}, \beta) = 1/(1 + e)^{-y(\{X_v\}, \beta)} \qquad (5)$$
where $y(\{X_v\}, \beta)$ = linear function + power function (section 2.3):
$$y(\{X_v\}, \beta) = \beta_0 + \Sigma_v(b_{0,i} \times X_v + b_{1,v} c^{X_v}) \qquad (6)$$

This works for single land points, where a location burns or does not burn. We extend this
concept to derive the Maximum Entropy solution for fractional burned area by integrating over
a larger grid cell area. Here we consider that when dividing a gridcell indefinitely, the subcell
sizes approach infinitesimally small values and the data within each subcell starts to behave
like continuous data. We adapted Eq. (3) and (4) to work with continuous data:
$$P(\beta \mid \{Obs_i\}, \{X_{iv}\}) \propto P(\beta) \times \Pi_i^n \Pi_j^s P(Obs_{ij} \mid \{X_{iv}\}, \beta)^{1/s} \qquad (7)$$
Where *n* is the observations sample size, *j* is the individual subgrid, and *s* is the subgrid sample
size.  If, for a given $Obs_i$, *m* of the *s* subgrid cells burn, then we can adapt Eq. (4) to get:




$$P(m/s \,|\, \{X_{iv}\}, \beta) \,=\, \Pi_j^s P(1 \,|\, \{X_{iv}\}, \beta)^{\,m} \,\times P(0 \,|\, \beta)^{s-m}$$

$$=\, f(\{X_{iv}\}, \beta)^m \,\times\, (1 - f(\{X_{iv}\}, \beta))^{m-s} \qquad (8)$$

and therefore:
$$P(\beta \,|\, \{m_i/s_i\}, \{X_{iv}\}) \,\propto\, P(\beta) \,\times\, \Pi_i^n f(\{X_{iv}\}, \beta)^{m/s} \,\times\, (1 - f(\{X_{iv}\}, \beta))^{(m-s)/s} \qquad (9)$$

When $s \rightarrow \infty$, $m/s$ becomes burned area fraction (BF). Then:
$$P(\beta \,|\, \{BF_i\}, \{X_{iv}\}) \,\propto\, P(\beta) \,\times\, \Pi_i^n f(\{X_{iv}\}, \beta)^{BF_i} \,\times\, (1 - f(\{X_{iv}\}, \beta))^{1-BF_i} \qquad (10)$$

This solution assumes that burning conditions at a specific location solely explain the
likelihood of burning. In reality, fires spread and, particularly at higher burned areas, they may
overlap. We, therefore, modify $Obs_i$ so that it represents what the burned fraction of a gridcell
would looks like if it was one large fire with no overlapping burning:
$$Obs_i \,=\, Obs_{i,0} \,\times\, (1 + Q) \,/(Obs_{i,0} \,\times\, Q \,+\, 1) \qquad (11)$$

Where $Obs_{i,0}$ is the true observation, and $Q$ is a modifier parameter to remove the effects of
fire overlap.
Lastly, to account for variations in land cover to assign between natural and non-natural
vegetation, which can be very small in some cells, we introduced a weighting factor $w$ when
assessing fire categories. This weighting factor considers the individual area of each grid cell,
ensuring that cells with smaller vegetation cover contribute proportionally to the analysis, as
in Eq. 12 below:
$$P(\beta \,|\, \{BF\}, \{X_{iv}\}) \,\propto\, P(\beta) \,\times\, \Pi_i^n f(\{X_{iv}\}, \beta)^{BF_i \times w} \,\times\, (1 - f(\{X_{iv}\}, \beta))^{(1-BF) \times w} \quad (12)$$

We use weak, uninformed prior distributions for our Eq. (6) parameters. $\beta_0$, $b_{0,i}$ and $b_{1,i}$ priors
were set as a normal distribution with a mean of 0 and a standard deviation of 100, and $c$ a
lognormal with a $\mu$ of 0 and a $\sigma$ of 1.
**2.5 Model evaluation**
The model's main goal is to accurately quantify uncertainties, which we tested by analyzing
where the observations fell in the model's posterior probability distribution (Eq. 10). If more





than 20% of the observations fall outside the 10th-90th percentile range, the uncertainty range
is too narrow. Conversely, if observations cluster around 50%, the uncertainty range is too
wide. We aim to minimize uncertainty constraints without compromising accuracy. When
evaluating the model against 2010-2019 observations, we also investigated how likely the
observations are given the optimized model (P(Observed|Simulated)), as per Kelley et al.
(2021). Using a different time period from the optimization, we ensure an independent model
evaluation. If the out-of-sample observations are more likely given the model, then the model
performs well. We use a likelihood of 50% to indicate adequate performance.

We calculate the probability of an observation given our model by integrating the observation's
likelihood across parameter space, weighted by the parameter likelihood given our training in
section 2.4:
$$P(Y|(X,\beta|\{BF_0\},\{X_0\})) = \int_\beta P(\beta|\{BF_i\}) \times P(Y|\beta)\, d\beta \qquad (13)$$


which, combined with Eq. (10), gives us:

$$P(Y|(X,\beta|\{BF_0\},\{X_0\})) = \int_\beta P(\beta|\{BF_i\}) \times f(X,\beta)^Y \times (1 - f(X,\beta))^{1-Y} \quad (14)$$


Where Y is an observation and X corresponds to the model inputs at the time and location of
Y. We approximate this by sampling 200 parameter ensemble members from each of our five
chains, providing us with 1000 ensemble members. The frequency of these 1000 in parameter
gives us "$P(\beta|\{BF_i\})$" in Eq. (14). We then drive the model with each parameter combination
to give us $f(X,\beta)$. We used the iris package (MET OFFICE, 2023) with Python version 3
(Python Software Foundation, https://www.python.org/) for sampling.

We also determined the percentile of our observations within the model's posterior probability
distribution. In an unbiased model, we expect the observation position to be essentially random,
with the mean over many samples tending towards the middle of the distribution (i.e., a
percentile of 50%). We mapped out the mean position of the observations for the 30 time steps
(3 months, August, September, October, for 10 years) tested (Fig. 6). The p-value in Fig. 7
uses the student t-test to ascertain if the mean of the posterior position of the monthly
observations for a given gridcell (mean bias) is significantly different 50% (i.e, the model is
biased). A mean bias near 0 indicates that observations are consistently smaller than the



simulations, and near 1 indicates that the observations are greater than the simulations. Low p-
numbers indicate where the model is biased towards a probability distribution, which tends to
suggest too low or high burning.

**2.6 Variables analysis**

We assessed the behavior of the variables against the burned area simulations by generating
response maps for our variable groups in a similar way to Kelley et al. (2019). In the potential
maps, we set each variable in the group to their median and kept the others at their original
values. The median, representing the middle value in a dataset, was chosen because it is less
affected by extreme values compared to the mean. The maps were subtracted from the original
simulations (control - potential response) to quantify the influence of the target group on the
model's response. This approach enables the assessment of burned area response when the
variable deviates from the median and assumes its original values. The agreement maps for the
potential response are then the percentage of the modeled distribution that shows an increase
in burning in each Biome. To compute the sensitivity response, we took the difference between
a simulation where we subtracted 0.05 and added 0.05 fraction of the training range of the
variable of interest. The goal was to understand how burned area responds to marginal
variations in the variables.

**3 RESULTS**

We present the results in two sections. The first section focuses on the model's performance in
simulating the observations, while the second section delves into the simulation's response to
the predictor variables.

**3.1 Model simulations and performance**

We performed simulations of burned area across each Brazilian biome and fire category, and
the resulting maps are shown in Fig. 5. The three simulation runs (ALL, NAT, and NON)
successfully captured uncertainties in all Biomes, with most observations falling within the
10th to 90th percentiles of the model. However, the model exhibits variations in uncertainties
based on the simulation category. For instance, in Amazonia, a biome characterized by a vast
expanse of natural vegetation, uncertainties were smaller in NAT simulations, contrasting with
larger uncertainties observed in NON-simulations, especially in areas where observed burned




areas are small or zero (Fig. 5). Similarly, the Pantanal displayed lower uncertainties in NAT
simulations, with values reaching up to 10%, while NON simulations registered uncertainties
up to 20% of burned area. The Atlantic Forest, a biome distinguished by non-natural vegetation,
exhibited smaller uncertainties in NON simulations. These findings indicate that the
segregation of fire categories (ALL/NAT/NON) substantially impacts the model's response.
Conversely, the model struggles to accurately capture large burned areas (> 10%) in central
regions of Brazil across all three simulations, mostly where the Cerrado biome is located.

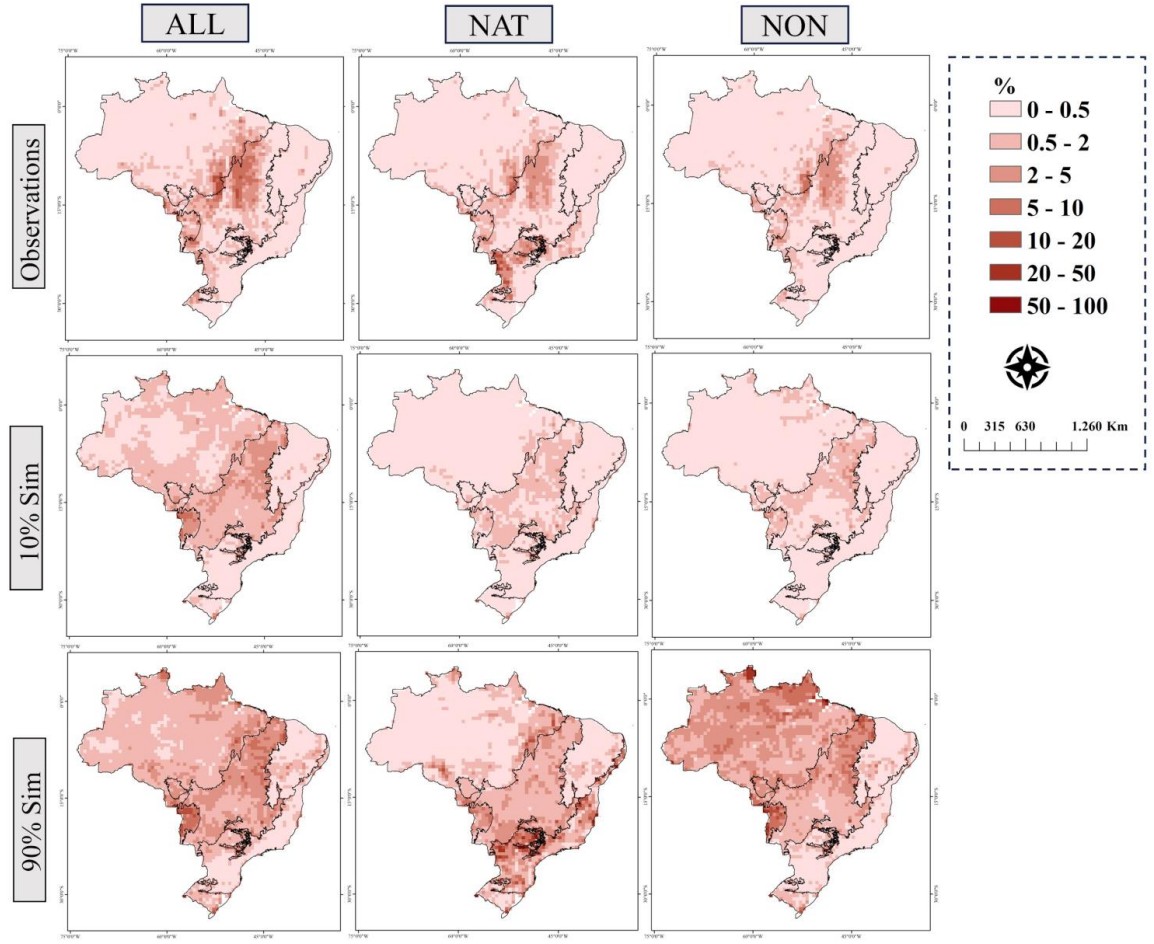

**Figure 5: Maps of modeled and observed % burned area. First row: observed burned area,**
**July-September 2002-2009 annual average for ALL (left), NAT (middle) and NON (right).**
**Second and third row: as top row but simulated by the model 10th and 90th percentiles,**
**respectively.**





In Bayesian inference, the likelihood expresses the probability of observing a particular event
given the model's parameters. Our results imply a strong agreement between the parameters of
the model and the observations (Table 2), even during the months when the observations were
less likely. The mean likelihood during these months was above 90% across all Biomes in all
simulations, except for the Pantanal, where the likelihood was lower (78% for ALL and 87%
for NON) but still satisfactory. The percentiles indicated that in the Pantanal, the likelihood of
the observations for ALL varied between 59% to 91%. In contrast, other Biomes presented a
minimum likelihood of 80%. During months of best performance, most biomes aligned with
the observations, achieving its maximum likelihood (100%) on average. The Pantanal,
however, presented the lowest values, with 97% for both ALL and NON simulations.

**Table 2.** Likelihood (%) per biome of the observations given the model parameters over all
cells and timesteps. 10% (left) indicates months/cells with worst performance, while 90%
(right) indicates best performance.

| Biome | Worst performance | | | Best performance | | |
|---|---|---|---|---|---|---|
| | Mean | 10th percentile | 90th percentile | Mean | 10th Percentile | 90th Percentile |
| **Likelihood - All fires** | | | | | | |
| Amazon | 95 | 89 | 99 | 99 | 98 | 100 |
| Caatinga | 99 | 98 | 100 | 100 | 100 | 100 |
| Cerrado | 90 | 80 | 97 | 99 | 98 | 100 |
| Atlantic Forest | 99 | 97 | 100 | 100 | 100 | 100 |
| Pampa | 96 | 92 | 100 | 99 | 98 | 100 |
| Pantanal | 78 | 59 | 91 | 97 | 93 | 100 |
| **Natural** | | | | | | |
| Amazon | 98 | 95 | 100 | 100 | 100 | 100 |
| Caatinga | 99 | 99 | 100 | 100 | 100 | 100 |
| Cerrado | 95 | 91 | 99 | 100 | 99 | 100 |
| Atlantic Forest | 99 | 98 | 100 | 100 | 100 | 100 |
| Pampa | 97 | 95 | 100 | 99 | 98 | 100 |
| Pantanal | 92 | 86 | 98 | 100 | 99 | 100 |
| **Non - natural** | | | | | | |
| Amazon | 95 | 91 | 99 | 99 | 98 | 100 |
| Caatinga | 99 | 99 | 100 | 100 | 100 | 100 |
| Cerrado | 94 | 88 | 99 | 99 | 98 | 100 |
| Atlantic Forest | 99 | 98 | 100 | 100 | 100 | 100 |
| Pampa | 97 | 94 | 100 | 99 | 98 | 100 |
| Pantanal | 87 | 78 | 96 | 97 | 93 | 100 |


Figure 6 presents the likelihood per pixel. Areas without values indicate zones where burned
area is zero, making the likelihood calculations inapplicable. The spatial likelihood analysis



provides additional insights into the model's robustness across different biomes and fire
categories. The results underscore the model's effective performance across the biomes.
Notably, the likelihood remained very high for the Atlantic Forest, Caatinga, and Pampa
biomes even in the months and locations where observations were less likely. A high likelihood
is also observed for NAT in Amazonia, except for the south and east, which contain most of
the non-natural vegetation. Lower performance is evident in the simulations for both ALL and
NON in this Biome, indicating that stratifyingfire categories by vegetation type could be a good
strategy to enhance model performance in Amazonia, or isolating fire categories where the
model has higher predictive ability. Similarly, the Pantanal showed the best performance for
NAT, but lower performance for ALL and NAT across the majority of the Biome. In contrast,
Cerrado performed better than most biomes for NON during the months of worst performance.

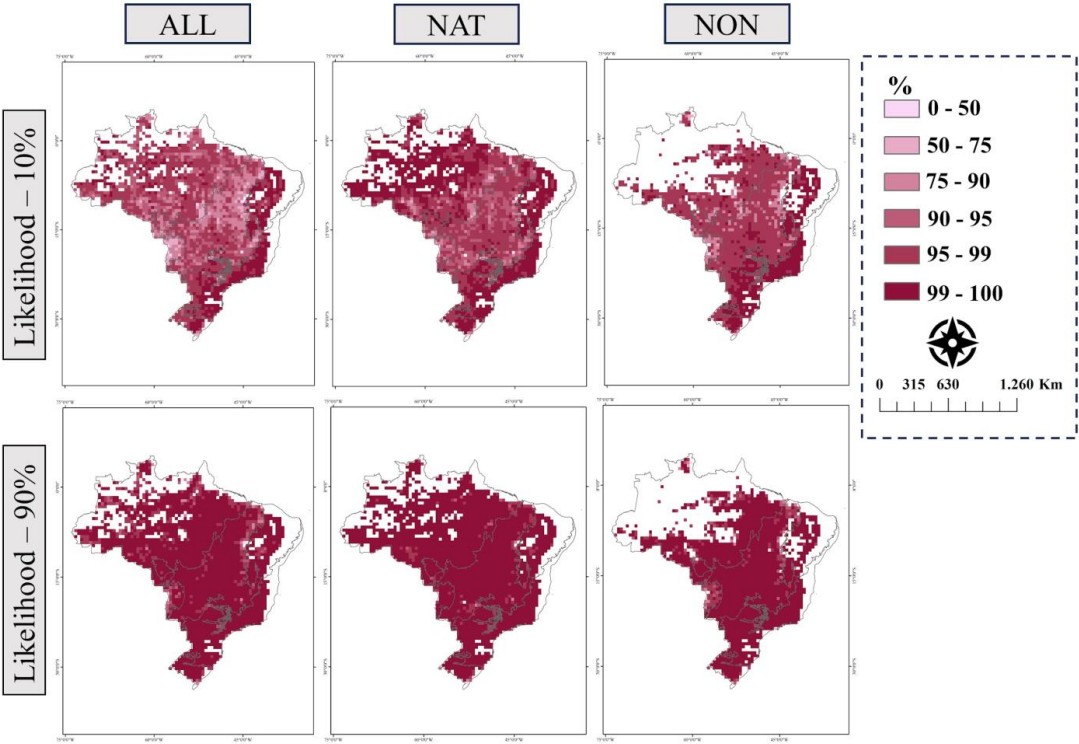

**Figure 6: Spatial likelihood of the observations given the model parameters considering the
months with worst performance (top row) and the months with best performance (bottom
row). A satisfactory performance of the model is considered with values above 0.5.**






Despite the high likelihood associated with the observations, the model simulations exhibit a
certain degree of bias across the three categories. A mean bias near 0.5 indicates no bias, as the
observations fall in the middle of the model's distribution. Amazonia and Cerrado showed mean
biases of 0.28 and 0.29 for ALL respectively, indicating an overestimation by the simulations
at lower burned areas. The Atlantic Forest presented a mean bias of 0.51, suggesting that,
overall, the model is unbiased although some pixels may still be biased. Similarly, Pampa
(0.42) and Caatinga (0.61) showed values near 0.5, indicating a lower degree of bias. In
contrast, a mean bias of 0.17 in the Pantanal suggests an overestimation of burned area by the
model, especially at lower levels. However, the model can distinguish between lower and high
burned areas in Pantanal (Fig. 5), indicating its ability to identify periods and locations of more
extreme burning, even if it does not exactly capture the correct magnitude.

Generally, higher uncertainties are observed for NAT and NON simulations, but a notable
improvement in bias is evident when compared to the ALL simulations. In the NAT
simulations, the model achieved its most favorable outcomes in Pampa (0.53) and Amazonia
(0.40), with the Pantanal also showing a noticeable improvement (0.34). The biases of 0.74 in
Caatinga and 0.72 in the Atlantic Forest indicate a trend toward underestimation in this fire
category. In Cerrado, a bias value of 0.33 was observed for NAT, aligning with the pattern seen
in the ALL simulations and suggesting a consistent overestimation, particularly for lower
burned areas.

In the NON simulations, Amazonia exhibited a bias of 0.38 but overestimated lower burned
areas. Cerrado and Pantanal showed similar patterns to those in the NAT simulations, with
respective mean biases of 0.36 and 0.31. The model tended to underestimate burned areas in
the Caatinga (0.81), particularly at higher burned areas. While Atlantic Forest (0.58) and Pampa
(0.59) showcased the most unbiased simulations for the NAT fire category, slight
underestimation of burned areas were noted in some instances (Fig. 7).

The spatial distribution of the mean bias, as depicted in Fig. 7, exhibits considerable variation.
Pixels without values indicate zero burned area in the observations, where, by definition, the
observation will always fall at the 0th percentile of the model posterior distribution.
Consequently, the bias metric does not provide meaningful information for these pixels. The
p-values reveal that in numerous areas, the bias is not statistically different from 0.5 (p-value



> 0.05; indicated by brown color), suggesting unbiased simulations in these regions.
Specifically, lower fires in Amazonia tend to occur in areas of natural vegetation, where NAT
simulations exhibit a non-significant bias. In these regions, ALL simulations tend to
overestimate burned area. In southeastern Amazonia, fires were underestimated across all three
fire categories, especially for NAT.

In Caatinga, all three simulations exhibited similar performance, significantly underestimating
fires, particularly in the northern part of the Biome. The Atlantic Forest displayed better results
for both ALL and NON, with a substantial area exhibiting non-significant bias. The fragmented
landscape of this Biome likely limits data availability for NAT, possibly explaining the lower
performance in this fire category. In contrast, Cerrado demonstrated a consistent pattern across
all three fire categories, predominantly overestimating fires, especially in the south and
northeast. While some underestimation occurred in the central biome, it was mostly non-
significant. In Pantanal, the simulation consistently overestimated burned area across all three
categories, with ALL simulations showing significant overestimation throughout the Biome.
Finally, Pampa displayed a non-significant bias across most of the region, except for the
northwest, where the model underestimated burning in all three simulations.

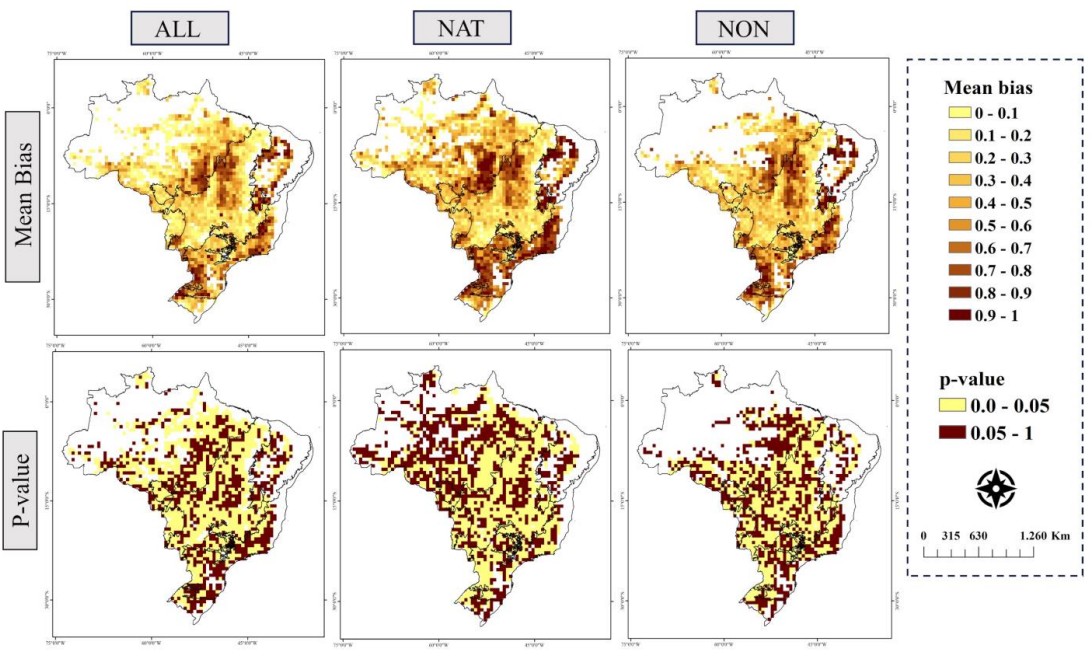



**Figure 7: Top row: Spatial mean bias of the modeled burned area to ALL (left), NAT**
**(middle) and NON (right). Bottom row: Significance of the mean bias considering a 95%**
**confidence level (p-value < 0 .05). Pixels with p-value > 0.05 (brown color) are not**
**significantly different from 0.5 mean bias meaning that they are unbiased.**

**3.2 Response of the modeled burned area to the explanatory variables**

We assessed the potential and Sensitivity responses of the variables (Fig. 8, 9 and 10). The
potential response offers insights into changes in burned area when variables deviate from the
median, thereby identifying areas where responses tend to drive or suppress burning. In
contrast, the sensitivity response provides information on how marginal changes in variables
affect burned area (KELLEY et al. 2019). Together, these analyses highlight areas susceptible
to more extreme burning (i.e., where the burned area is sensitive to variables that tend to cause
higher potential burning).

For ALL burned area (Fig. 8), variations of Group 1 (Maximum Temperature and Precipitation)
from the median is very likely to lead to an increase in the burned area in 62.33% of Amazonia
(with a likelihood of over 80%). This means that when these variables assume their actual
values in this Biome, the burned area tends to be higher, with increases up to 1% in the western
edge and 10% in the north, northeastern and southeastern of the biome. Conversely, these
variations contributed to a reduced burned area in 33.57% of Amazonia, predominantly
observed in the western and central areas, suggesting that Maximum Temperature and
Precipitation tend to suppress burned area in these regions. In 4.08% of the biome, the influence
of Group 1 variables on burned areas is not confidently predictable in terms of whether they
will lead to an increase or decrease (with likelihood between 40% and 60%), and the model
showed strong confidence only in the regions where these variables are major drivers and
suppressors of burning. Our results indicate that the entire Amazon is highly sensitive to minor
variations in Group 1 variables for ALL (Fig. 8). Nonetheless, the middle and western regions
tended to be up to three times less sensitive than the rest of the biome.

In the Atlantic Forest, approximately 63.33% of the biome will likely experience an increase
in burned areas when Temperature and Precipitation assume their real vs median values, mostly
limited to 1% extra burning. This small increase highlights that these drivers do not have a
major influence on driving high levels of total burned area. Reduction of burned area is



observed in the western portion, encompassing 31.79% of the biome. Uncertainties linked to
Group 1 variables were found in 4.87% of the Atlantic Forest. Moreover, this biome showed
an overall lower sensitivity to climate.

In Cerrado, Group 1 is likely to drive burned area up to 6% in 58.30% of the biome, primarily
in the eastern part. Conversely, 37.16% of Cerrado is expected to observe a reduced burned
area by up to 10%, showing quite a range in the influence in mean burned area from the variable
group. The remaining 4.53% of the area remains uncertain. Cerrado exhibited high sensitivity
to changes in Group 1, except for the central region of the biome, which showed comparatively
lower sensitivity. In the Pantanal, the central and northern areas are likely to experience an
increase in burned area by up to 1% due to variations in Group 1, accounting for 51.92% of
their total area. Conversely, the borders of the Pantanal, particularly the south, exhibited a
reduction in burned area (42.30% of the Pantanal). Approximately 5.76% of the Pantanal
landscape remains uncertain regarding the direction of changes. The entire biome presented
considerable sensitivity for small variations in Group 1. Pampa exhibited a high likelihood of
increased burned area in 70.14% of the region, mainly limited to 1%. We found a high
likelihood of reduction in 26.86% of Pampa, located in the northwestern, and in 2.98% of the
biome it is unclear the direction of changes. Pampa's west and southeastern edges showed to
be more sensitive to Group 1. The southern and eastern portions of Caatinga are likely to face
an increase in burned area by up to 4%, affecting 51.23% of the biome, attributable to the
influence of Group 1. Conversely, 47.34% of Caatinga, particularly in the northern and western,
is more likely that the burned area will diminish, while 1.41% is unclear. In general, the biome
showed less sensitivity to Group 1, with slightly higher sensitivities observed in the central and
northeast of the biome.

For Group 2 variables (Edge Density and Road Density), 47.37% of Amazonia will likely
experience an increase in burned area when these variables deviate from the median. This
increase is predominantly limited to 1%, concentrated in the western, central, and northeast
regions. Conversely, areas with higher edge and road densities show a reduced burned area of
up to 11%, covering 51.82% of Amazonia. This is a 12% range in burned area, substantial for
a fire-sensitive biome. Overall, the biome displays moderate sensitivity to minor variations in
Group 2, with higher sensitivity observed along its borders. The response in the Atlantic Forest
exhibited more uncertainty in the 10th and 90th percentiles. Still, the likelihood indicates that
42.30% of the biome will likely experience increased burned areas of up to 2%, primarily





located in the north and eastern edges. Small reductions are found in 54.87% of the biome,
limited to 0.2%. Regions where increases are more likely also demonstrate greater sensitivity
to Group 2, showing the potential for these drivers to have a disproportionate influence on
extreme levels of burning.

The Cerrado biome exhibited high spatial variability in response to Group 2, with a nearly
equal mix of pixels where an increase (47.28%) and decrease (44.56%) in burned area is more
likely to occur, both limited to 2.5%. The northeast of the biome displayed higher sensitivity
to Group 2. In Pantanal, the central and southern regions are more likely to experience a
decreased burned area, encompassing 53.84% of the biome. However, an increase is found in
42.30% of Pantanal, limited to 8%. The Pantanal demonstrated sensitivity to Group 2,
especially in the north. In Pampa, 47.76% of the region exhibited increased burned areas, while
reductions occur in 47% of it. Increases reached up to 4%, primarily in the western portion.
These regions where an increase is likely also showed higher sensitivities. In Caatinga, a
reduction in burned area is likely to occur in 50.17% of the biome, while an increase is expected
in 38.86% of it. Approximately 10.95% of the biome remains uncertain about the direction of
change. In areas where results do suggest a confidence change, increases are mainly located in
the middle of the biome.

In the context of Group 3 variables (Forest, Pasture, and Carbon in dead vegetation),
approximately 53% of Amazonia will likely experience larger burned areas, primarily
concentrated in the arc of deforestation (along the southern and eastern edges of the Amazon),
reaching up to 10%. Conversely, reductions are observed in 42% of the biome, with 4.23%
remaining uncertain. While displaying less sensitivity to minor changes than other groups,
certain areas such as the cross borders with Cerrado and north exhibit higher sensitivity within
the biome. In the Atlantic Forest, increased burned areas are observed in 41.53% of the region,
while reductions are noted in 54.87%. Decreases in the biome are primarily observed in the
central southern and eastern areas, with magnitudes reaching up to 0.7%. Overall, the
sensitivity in this biome is lower although the spatial variation shows heightened sensitivity in
the 90th percentile for some pixels across the biome.

In the Cerrado biome, burning in the middle south and northeast edges is not likely driven by
Group 3 variables, covering 54.83% of the biome. Conversely, the north, northeast, and part of
the south (39.72% of Cerrado) may experience increased burned areas of up to 10%. Regions



with higher likelihood of increase also demonstrate greater sensitivity to small variations in
Group 3. Pantanal shows approximately 30.77% of its area likely to experience up to a 10%
increase in burned areas, mainly in the north and southeastern regions. Conversely, edges and
the southern part are more prone to reductions, encompassing 55.76% of the biome, while 13%
remain uncertain. Pantanal demonstrates high sensitivity overall to Group 3. In Pampas,
52.23% of the region is more likely to see increased burned areas of up to 3.5%, while
reductions are observed in 44.77% of the area. The western part and eastern edges of the biome
show greater sensitivity to minor changes in Group 3. In Caatinga, approximately 53.35% of
the biome is likely to experience reduced burned area while 38.16% is likely to see up to 3%
increases. The central and northeast regions, where increases are expected, also exhibit higher
sensitivity to minor shifts in Group 3.

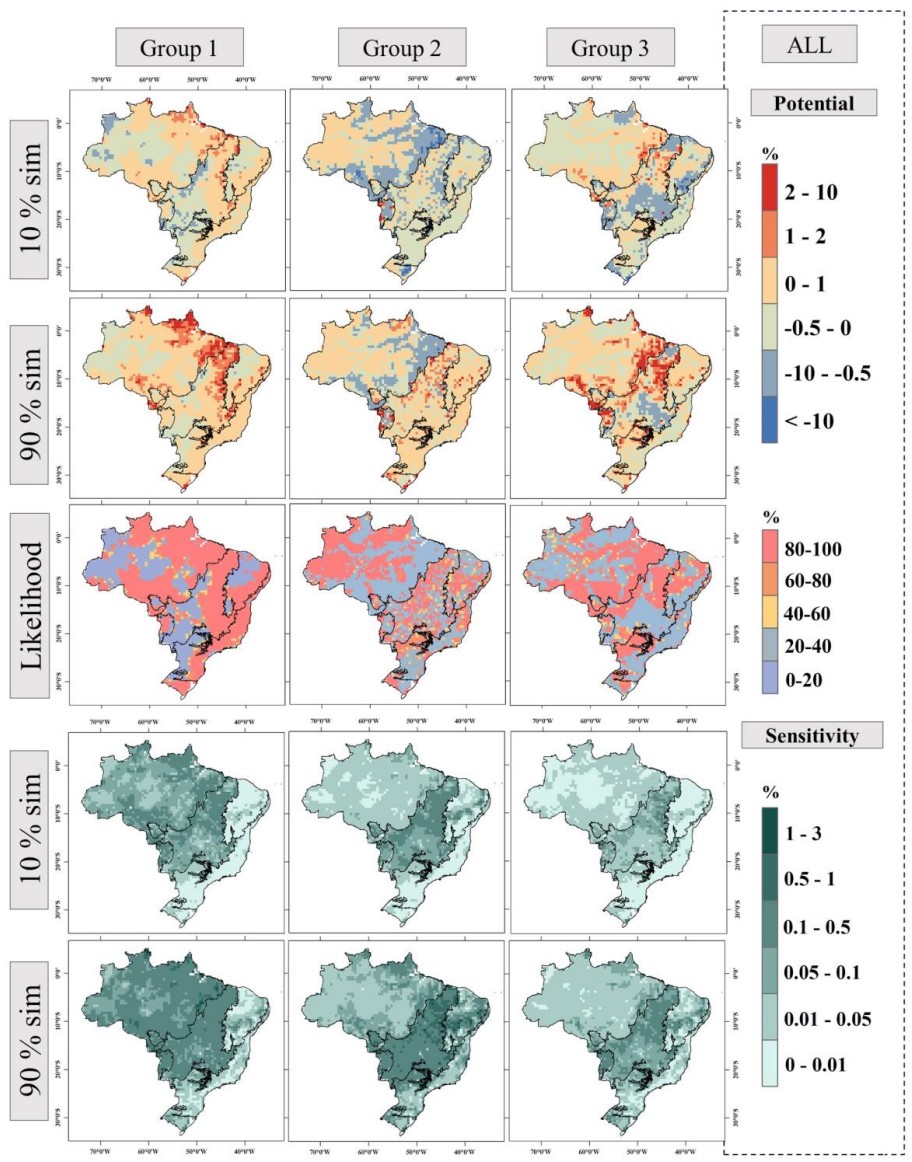


**Figure 8: Response maps to ALL displaying the potential 10h percentile (first row), 90th percentile (second row), likelihood (third row) and sensitivity responses 10th percentile (fourth row) and 90th percentile (fifth rows). Each column presents the results for one group of variables.**


Similar spatial patterns to ALL were observed for NAT when considering Group 1 across all

biomes (Fig. 9). In the Amazon, Group 1 will likely increase burned area in 63.79% of the

biome. Reductions are found in 29.92%, while 6.27% display an unclear response. This

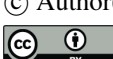



indicates a 2% increase in areas with uncertain responses, particularly in the southeastern
region of the Amazon. Sensitivity analysis reveals that the borders of the Amazon are more
sensitive to Group 1, whereas areas with forest cover < 83% (Fig. 3) exhibit lower sensitivity.
In the Atlantic Forest, Group 1 is likely to drive burned area changes in 67.95% of the biome.
Conversely, 19.23% is likely unaffected by Group 1, with 12.82% remaining unclear,
representing an 8% increase compared to ALL. The Sensitivity to Group 1 was similar to ALL,
generally lower for this biome.

In Cerrado, Group 1 contributes to increased burned area in 61.78% of the biome. However, in
32.78% of the area, Group 1 is likely not a driving factor for the burned area, and in 4.53%,
the response is unclear. The biome also exhibits sensitivity to minor variations in Group 1 for
NAT, albeit slightly lower in some areas (Fig. 9) than ALL. In Pantanal, 80.76% of its area
likely has Group 1 as drivers of burned area in NAT, representing an increase of almost 30%
compared to ALL. Areas not influenced by this group decreased by 25% compared to ALL
(15.38% of Pantanal), while 3.84% remains unclear. The sensitivity analysis closely resembled
ALL, with the entire biome significantly responding to variations in Group 1. In Pampas, it is
likely that variations from the median lead to increased burning in 70.14% of the biome.
Sensitivity is similar to ALL, primarily in the west but generally lower. Caatinga follows a
similar pattern to ALL, with Group 1 influencing burning in 48.76% of the biome. Uncertainty
increased to 4.94% of the biome, and sensitivity is similar, affecting mainly the middle and
northeast regions.

For Group 2, Amazon presented a more uncertain response between the 10th and 90th
percentiles. However, the likelihood showed a marked pattern very similar to ALL where
47.37% of the biome has Group 2 as a driver of burning. Similar to Group 1, the sensitivity
was lower in highly forested areas. For NAT, the Atlantic Forest showed large areas with an
unclear response (Fig. 9), covering 41.79% of the biome. The areas where burning is likely to
be driven by Group 2 encompasses 26.41%, a reduction of 15% when compared to ALL. The
sensitivity was similar to ALL, with slightly higher values in some pixels. The Cerrado showed
variation within the biome, with 45.61% of its area identified as potentially driven by Group 2
in NAT. While the sensitivity was lower than in ALL, it remained significant within Cerrado.
Pantanal exhibited Group 2 as a driver of burning in 46.15% of the biome, displaying a spatial
pattern for the likelihood very similar to ALL. However, sensitivity was lower in the middle
of Pantanal compared to the North and edges. Similarly, Pampa presented a response similar



for both potential and sensitivity as in ALL, with 47.76% of areas likely to experience increased
burning driven by Group 2. In Caatinga, areas likely to experience increased burning accounted
for 37.45% of the biome, and the regions with unclear responses were 6.72% higher than in
ALL (17.67%). Sensitivity showed the same pattern as in ALL.

Amazonia showed a 4% increase in areas with unclear responses for Group 3 to 8.10%
compared to ALL. Regions susceptible to burning due to this group totaled 54.74% of the
biome. Densely forested areas also exhibited lower sensitivity to minor shifts in Group 3. In
Atlantic Forest, Group 3 is likely to be a driver of burned area in 41.02% of the biome, very
similar to ALL (41.53%). Similarly, the sensitivity followed the spatial pattern of ALL with an
overall lower sensitivity presenting slightly higher in some pixels. Areas prone to burning in
the Cerrado due to Group 3 reduced by 10.84%, totaling 43.95% compared to ALL. The
reduction was concentrated in the northeast, while in the southwest there was an increase in
the likelihood of burning due to Group 3. The sensitivity reduced in the northeast, varying
across the biome. Within the Pantanal, regions susceptible to burning due to Group 3 comprised
32.69% of the area. Regions with an unclear response increased by 4.30%, encompassing
17.30% of the region and concentrated in the eastern edges.

In Pampas, 44.77% of the biome is likely to burn due to Group 3, while 17.30% of the biomes
showed an unclear response. The sensitivity pattern for NAT followed ALL, concentrated in
the western and eastern edges. The Caatinga accounted for 35.68% of areas prone to burning,
with higher sensitivities observed in the middle and eastern regions of the biome.









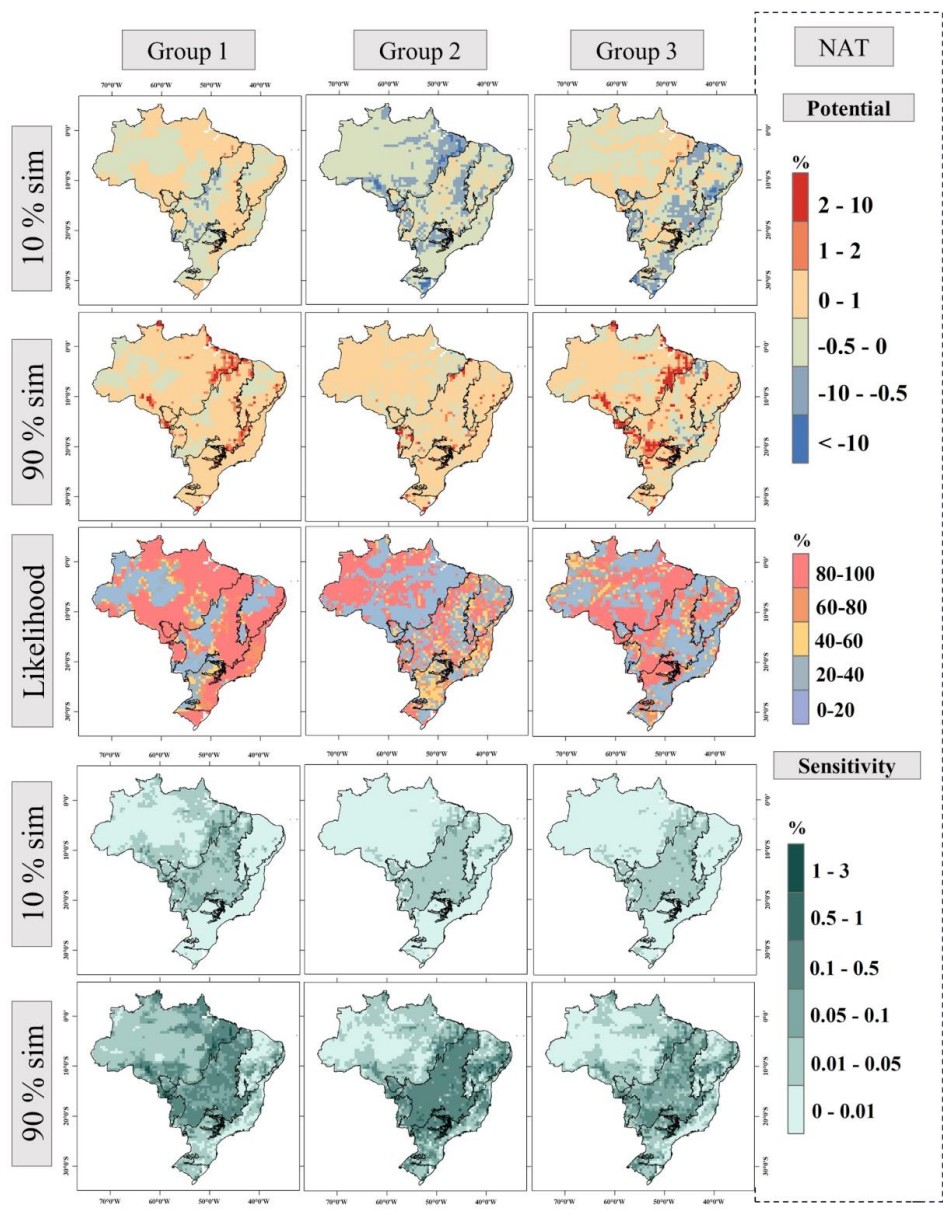

**Figure 9: Same as Fig. 9 but for NAT.**

Higher uncertainties were found in the potential response for NON, meaning that the range of possible outcomes was generally larger for this category (Fig. 10). However, the likelihood showed similar spatial variation, although unclear responses increased. Group 1 acts as a driver of burning in 62.99% of Amazonia, a similar number when compared to NAT and ALL. The



main difference for this category is the magnitude of increase, which is higher at the edges and
in the middle of the biome. Likewise, the sensitivity was higher, especially in the 90th
percentile. The potential and sensitivity response of the Atlantic Forest was quite similar for
the three categories, with 64.61% likely to have Group 1 increasing burning in the biome.
Within the Cerrado, a 13.15% and 9.67% increase in areas susceptible to burning is observed
compared to ALL and NAT respectively (totaling 71.45%). Unclear responses were higher and
reached 9.21% of the biome. Sensitivity was higher in the northeast of the biome. For Pantanal,
NON comprised 69.23% of areas likely to burn due to Group 1. An increase in unclear
responses of 7.7% and 9.62% compared to ALL and NAT respectively was found (totaling
13.45% of the biome). The magnitude of increase was also higher for NON. Sensitivity levels
were mostly high across the biome. Within Pampas, 79.10% of the biome was considered likely
to burn due to Group 1. The sensitivity was larger at the edges of the biome. The potential and
sensitivity responses of Caatinga followed a similar pattern between the categories, where
47.70% of the biome is likely to be susceptible to burning due to Group 1.

Similarly, the main difference for Group 2 in Amazonia was the increase, which reached up to
10% in the North and middle of the biome. Most of the biome shows high sensitivity. Within
the Atlantic Forest, there was a notable reduction of 30.51% in regions with unclear responses
compared to the NAT, where the proportion was 11.28%. Regions likely to increase burned
area due to fragmentation comprise 41.28% of the biome, an increase of 14.87% compared to
NAT. Sensitivity showed a similar pattern for the three categories where regions likely to
increase burning presented higher sensitivities. In Cerrado, approximately 41.54% of its area
is likely susceptible to increased burning due to fragmentation, with 15.70% exhibiting unclear
responses. Higher sensitivity was observed in the northeastern region of the biome. Pantanal
showed a 40.38% likely increase and a significant sensitivity across the biome. Pampas patterns
for potential and sensitivity responses were similar to ALL and NAT, with 49.25% of the biome
likely to increase burning. However, the likelihood was comparatively lower (between 60%
and 80%). For Caatinga, it is likely to increase burning in 36.39% of the biome, while the
regions with unclear response reached 21.90%. Sensitivity displayed a similar pattern to ALL
and NAT with higher sensitivities in the middle and northeast.

Group 3 exhibited higher uncertainties in Amazonia between the 10th and 90th percentiles.
The likelihood of increase encompasses 44.59% of the biome, while areas with unclear
responses surpass ALL and NAT, comprising 10.21%. Sensitivity was also higher, especially



in the north of Amazonia. The Atlantic Forest showed a similar pattern compared to ALL and
NAT with 38.71% of its area likely to increase and generally lower sensitivity to this group.
Cerrado exhibited a marked pattern where burning in the north is likely driven by Group 3,
encompassing 40.78% of the biome. These regions also exhibited higher sensitivity to minor
variations in Group 3. Unclear responses were identified in 11.48% of the biome. This Group
exhibited the highest level of unclear response in the Pantanal, totaling 30.77%. Meanwhile,
regions with a likelihood of increased burning decreased to 25%. The sensitivity was generally
high across the biome. This group also showed to be highly uncertain in Pampas, with 55.22%
of the biome presenting unclear responses. The areas likely to increase burning comprised
23.88% of Pampa, a reduction of 28.35% and 20.89% compared to ALL and NAT,
respectively. The sensitivity was similar in the three categories with slightly higher sensitivity
in the middle for NON. The Caatinga region exhibited a 35.33% portion of its area with a
heightened likelihood of increased burning attributed to Group 3, displaying a similar pattern
across all three categories concerning potential and sensitivity response.





**Figure 10: Same as Fig.9 but for NON.**





## 4 DISCUSSION

### 4.1 FLAME's performance in context

Our proposed model uniquely combines two previously distinct approaches employed in fire modeling: Bayesian inference and Maximum Entropy (KELLEY et al., 2021; FERREIRA et al., 2023). This combination allows for a more comprehensive understanding of fire dynamics as it models a probability distribution rather than singular values, a departure from conventional models (e.g. HANTSON et al., 2016; RABIN et al., 2017). Notably, our approach employs Maximum Entropy to capture the most uncertain outcomes that align with our priors, reflecting the stochastic nature of real-world fires. This concept contributes to a more nuanced and realistic representation of fire behavior. We conducted our analysis by categorizing the burned area into three categories: fires in both natural and non-natural vegetation (ALL), fires reaching natural vegetation (NAT), and fires reaching non-natural vegetation (NON). This classification yielded distinct results for each category with an overall improvement across the biomes for the NAT and NON. Moreover, this approach allows us to make more targeted conclusions.

The results demonstrate the robust performance of our model in capturing observations while providing a range of possible outcomes represented by the 10th and 90th percentiles. It is noteworthy that the model was capable of reproducing the observations in Pampa, Atlantic Forest and Caatinga, as these are areas where other methods used in previous studies have not performed well (NOGUEIRA et al. 2017, OLIVEIRA et al., 2022). Despite some level of bias in the results, even during periods of suboptimal performance, the likelihood of the observations remained consistently high, with the majority exceeding 80%. The Pantanal biome presented an exception, displaying a likelihood of 59% for the combined category (ALL), with improvement for specific categories, reaching 86% for NAT and 78% for NON. This biome encompasses a mosaic of vegetation types characterized by seasonally flooded areas which plays an important role on the fire dynamics of the region (DAMASCENO-JUNIOR et al.,2021). Fire in these areas were not included in this study due to our general approach, posing a limitation for simulation within this biome. However, our framework's adaptability means that future work could look at different explanatory variables, relationship variables and fire categorizations that could target performance in places like the Pantanal.



The MaxEnt species distribution model, which uses the same Maximum Entropy concept applied here, became quite popular in fire modeling studies (e.g., FONSECA et al., 2017; BANERJEE, 2021; FERREIRA et al., 2023). However, the MaxEnt software provides default settings, based on average values which are likely to change according to species, study region and environmental data (PHILLIPS and DUDIK, 2008). Additionally, these current settings are estimated to result in excessively complex models, potentially leading to overfitting (RADOSAVLJEVIC and ANDERSON, 2013). When employing MaxEnt, it is crucial to utilize independent evaluation data (PETERSON et al., 2011) such as that used in the present study. However, many studies assess performance by randomly partitioning occurrence data into calibration and evaluation datasets (CHEN et al., 2015; GÖLTAS et al., 2024). This approach limits the ability to obtain reliable estimates of model performance, generality, and transferability. Finally, the area under the receiver operating characteristic (ROC) curve, commonly known as AUC, is widely used as a standard method to evaluate the accuracy of MaxEnt-based models. Nonetheless, this measure does not provide information about the spatial distribution of the model's performance (LOBO et al., 2007; JIMÉNEZ-VALVERDE, 2011) which also potentially masks the spatial variability of the explanatory variables contribution to the model.

Currently, global fire models incompletely reproduce the observed spatial patterns of burned area. We found that FLAME captures high burning events, albeit not with the exact magnitude observed. This ability presents an advantage compared to many global fire models. While global fire modeling provides useful information into broad-scale patterns and trends, they are mostly designed to estimate global mean burned area (HANTSON et al., 2016; BURTON and LAMPE et al., 2023). As a result, its applicability to regional scales such as the Brazilian biomes is inherently limited. Furthermore, these models are typically constructed based on assumptions regarding variable relationships, which may not hold true in all locations due to variations in environmental conditions, ecosystem dynamics, and human activities. However, Earth System Models integrate feedback mechanisms between burned areas and predictor variables, enabling the evaluation of inter-variable effects. FLAME is not designed to capture these feedbacks, underscoring the need for tailored methodologies to address specific research questions.

**4.2 Burning controls across the biomes**



We combined our variables into three groups to assess their compound effect on the burned
area. This is a similar approach to Kelley et al. (2019) who also used a Bayesian framework to
assess drivers of global fire regimes. Nonetheless, Kelley et al. (2019)  considered only linear
responses which is especially challenging when considering the varying responses across the
globe. Our results highlighted  the spatial variability of each variable group's influence on
burning within and between each biome. The potential response displayed similar spatial
likelihood variation between the ALL, NAT and NON categories. However, differences were
still observed, especially for the fire-dependent biomes (Cerrado and Pantanal). Overall, the
uncertainties were larger for the NON category, particularly for Pampas and Pantanal.

For example, Maximum Temperature and Precipitation (Group 1) are likely drivers of burning
in large portions of each biome during the fire peak, as demonstrated by the potential and
sensitivity results. Our results indicate that in highly forested areas in Amazonia, climate alone
does not control burning, suggesting that forests can potentially mitigate the effects of climate
in burned area. These regions showed up to three times less sensitivity to minor variations of
climate for NAT while ALL and NON displayed high sensitivity in the whole biome. However,
natural landscapes, especially forests, are highly susceptible during extreme weather conditions
(DOS REIS et al., 2021; BARBOSA et al., 2022). This suggests that projected climate change
could greatly increase the risk of Amazon forest fires (FLORES et al., 2024). Moreover, non-
natural vegetation in Amazonia is mainly concentrated in the arc of deforestation, reducing the
samples for this category in other parts of the Amazon and potentially influencing the model's
response. An opposite dynamic was found in Cerrado and Pantanal. Regions with large areas
of natural vegetation were more likely to be influenced by climate. These regions were more
sensitive to minor variations in climate for NON in Cerrado while the entire Pantanal displayed
similar sensitivity in the three categories. This aligns with prior research showing that fires in
Cerrado are linked with meteorological conditions, particularly rainfall and temperature
(NOGUEIRA et al., 2017; LIBONATI et al., 2022; LI et al., 2022). Similarly, in Pantanal, the
2020 fire season revealed the connections between meteorological conditions and increased
burning in the biome (BARBOSA et al., 2022; LIBONATI et al. 2022b) and again during the
2023 El Niño. Barbosa et al., (2022), reported that 84% of the 2020 record of fires in Pantanal
occurred in natural vegetation, with a 514% increase from average within forests. Despite being
a combination with land use, the precipitation and maximum temperature anomalies were
particularly high in 2020, contributing to the spread of fires into fire-sensitive vegetation.





Group 2 (Edge density and Road density) encompasses variables expected to have uncertain
response across the biomes. Within Cerrado, 40.63% of its area will likely decrease burned
area for NAT due to Group 2. A high density of forest edges has been associated with a higher
incidence of fires in forest ecosystems (ARMENTERAs et al., 2013; SILVA-JUNIOR et al.,
2022). However, fragmentation can also act as a barrier to fire spread, potentially reducing fire
occurrences (DRISCOLL et al., 2021). Rosan et al., (2022), revealed that in Cerrado,
fragmentation correlates with a decrease in burned area fraction, while in  Amazonia, it is
linked to an increase in burning. Nevertheless, we found a decrease in burning where edge
densities are concentrated in the Amazon. This could indicate that the edges of the Amazon are
reaching a level of fragmentation that fires are impeded from spreading, considering the
reduction of aboveground biomass near forest edges (NUMATA et al., 2017). However, further
research is needed to test this hypothesis.

Depending on the landscape, road densities can also exhibit contrasting relationships with fires.
While more fires are expected surrounding roads (ARMENTERAs et al., 2017), less fires are
expected with increased density due to urbanization. The Atlantic Forest is a very fragmented
biome with very high densities of natural edges and roads (Fig. 3). We found an uncertain
response for NAT in 41.79% of the Atlantic Forest and only 26.41% likely to increase. Singh
and Huang (2022) suggests that the fragmentation partly explains burned area variation in the
Atlantic Forest where small patches are more vulnerable to fires. The majority of Caatinga is
likely to decrease burning due to Group 2. However, the sensitivity was up to three times higher
in the middle and northeast, which is more likely to increase. Antongiovanni et al. (2020)
discussed that fires in Caatinga occur at all edge distances, although they are slightly more
frequent at fragment edges. Nonetheless, the limited amount of studies across the different
biomes addressing these relationships makes it harder to understand the related uncertainties.

Group 3 is likely to influence burning in 54.74% of Amazonia for NAT, particularly in the arc
of deforestation. This suggests that the combination of less forest, increased pasture and more
fuel (Fig. 3) increases burning in natural lands in Amazonia, corroborating previous findings
(SILVEIRA et al., 2020; SILVEIRA et al., 2022). The relationship in Pantanal and Pampa
showed that these variables increase burning in 32.69% (NAT) and 25% in Pantanal and
44.78% (NAT) and 23.88% (NON) for Pampas. The regions with unclear responses were the
highest for NON, 30.77% of Pantanal and 55.22% of Pampa. These biomes are characterized
by lower forest and pasture cover (Fig. 3) with fires and cattle ranching mainly linked to



grasslands (BARBOSA et al., 2022; FIDELIS et al., 2022; CHIARAVALLOTI et al., 2023).
Thus, incorporating grassland cover in the model will likely reveal further relationships
between burned area and LULC in these biomes. Caatinga showed increased sensitivity where
Group 3 is likely to increase burning, matching the area of influence of Group 2. This area is
associated with low forest cover and soil carbon and moderate pasture cover. Araújo et al.
(2012), observed that due to the intermittent and scattered characteristics of cattle ranching in
the Caatinga, fires tend to occur mainly in natural vegetation, characterized by large cover of
savanna vegetation. Although our study provides a general overview of burning dynamics in
the biomes, targeting variables is highly recommended in future studies, especially where fires
are poorly understood as in Caatinga.

### 915    4.3 FLAME potentialities

Further developments are recommended to improve FLAME's capabilities. Exploring and
incorporating better-informed and additional priors may constrain the variables' response
uncertainties. Utilizing alternative metrics to assess drivers, particularly those tailored to
specific biomes, could offer a more nuanced understanding of the influencing factors. It could
also help improve biases in biomes such as the Pantanal. Customizing variable selection based
on biome characteristics would also contribute to a more biome-focused and contextually
relevant analysis. Consideration of different fire categories show how the model could be used
in further research. For instance, a more detailed stratification could involve categorizing fires
into distinct groups such as forest, agricultural, and deforestation fires. While deforestation data
was not incorporated in this study, efforts should be made to integrate this valuable information
where possible. Furthermore, accounting for the varying proportions of natural and non-natural
lands within each pixel, as demonstrated in this study, provides a more accurate landscape
representation. This contributes to improved simulations where these areas are very small. In
addition, finer grids and the subdivision of the biomes may uncover local processes, though
eventually fire spread between fine-scales would need to be considered. This could be crucial
for understanding localized patterns and improving the model's predictive capabilities. Perilous
modeling attempts often parameterize on a large regional basis. However, our approach allows
for optimization on much smaller areas while still quantifying the confidence in the analysis.
FLAME is flexible enough to be used in various locations and, through targeted benchmarking,
holds the potential to evaluate extreme fires, inter-annual and seasonal variability of fires,
project future fires, and simulate other hazards. With appropriate adaptations and



enhancements, FLAME has the potential to evolve into a robust model capable of simulating
terrestrial impacts effectively.

**5 FINAL CONSIDERATIONS**

The self-reinforcing cycle between fires and climate change makes it fundamental to improve
fire simulations. An understanding of what drives fires is essential for devising mitigation and
adaptation strategies. However, it can be particularly challenging due to the intricate interplay
of various factors, especially in a diverse country like Brazil. We propose a novel approach for
simulating burned area in the Brazilian biomes that keeps assumptions at a minimum whilst
quantifying uncertainties. The model performs well in all biomes, and enables the assessment
of fire categories and the grouped effect of variables. Furthermore, conventional modeling
efforts often parameterize at a large scale. FLAME enables optimization in smaller areas while
still providing a means to quantify confidence in the analysis.

Climate is an important factor in burned area in all biomes. Despite several studies showing
this relationship, climate-related uncertainties had not been extensively quantified, a gap this
research fulfills. Groups 2 (road and edge densities) and 3 (forest, pasture and soil carbon) and
the NON category showed the highest uncertainties among the responses. This highlights the
challenge in modeling human-related factors. Pantanal, Cerrado, and Amazonia showed higher
sensitivity to minor variations in the variables. It is important to note that sensitivity is more
important where burning is already high, which is the case in these biomes (ALENCAR et al.,
2022). None of the groups drive huge changes in burned area in the Atlantic Forest, though as
it is fire-sensitive, it still can have a large impact. Uncertain responses compound the
complexity of burned area drivers as different variables interact uniquely within each biome.
The same vegetation type may show contrasting responses to the same drivers in different
locations. Therefore, no universal fire management policies will fit the whole country. In
particular, Caatinga, Atlantic Forest and Pampa require further investigation. Emphasizing
regional-scale analysis is crucial for decision-makers and fire management strategies, enabling
more informed and effective prevention of fires.

**CODE AVAILABILITY**

FLAME 1.0 model code is available at https://doi.org/10.5281/zenodo.13367375 (Barbosa et
al., 2024a).




**DATA AVAILABILITY**



The data supporting this study is available at the Zenodo repository:
https://doi.org/10.5281/zenodo.11491125 (Barbosa et al, 2024b).

**AUTHOR CONTRIBUTIONS**

**Conceptualization:** MLFB, DIK, CAB, LOA

**Data Curation:** MLFB, DIK, AB

**Formal Analysis:** MLFB, DIK

**Methodology:** MLFB, DIK, CAB

**Resources/Software:** MLFB, DIK, CAB

**Visualization:** MLFB

**Funding acquisition:** MLFB, LOA

**Supervision:** LOA, DIK, CAB

**Resources:** LOA, DIK, CAB

**Writing – Original Draft Preparation:** MLFB

**Writing – Review & Editing:** MLFB, IJMF, RMV, AB, PGM

All co-authors approved the draft

**COMPETING INTERESTS**

The authors declare that they have no conflict of interest.

**FUNDING ACKNOWLEDGMENTS**
DIK was supported by the Natural Environment Research Council as part of the LTSM2
TerraFIRMA project and NC-International programme [NE/X006247/1] delivering National
Capability. This work and its contributors (CAB, AB) were funded by the Met Office Climate
Science for Service Partnership (CSSP) Brazil project which is supported by the Department
for Science, Innovation & Technology (DSIT). LOA acknowledges support by the São Paulo
Research Foundation (FAPESP) (projects: 2021/07660-2 and 2020/16457-3) and by the
National Council for Scientific and Technological Development (CNPq) project 409531/2021-
9 and productivity scholarship (process: 314473/2020-3). MLFB and IJMF were supported by
the Coordination for the Improvement of Higher Education Personnel (CAPES), Finance Code
001. MLFB and PGM acknowledges support by the São Paulo Research Foundation (FAPESP)



(project: 2021/11940-0). RMV thanks the São Paulo Research Foundation (FAPESP) for grants
2020/06470-2 and 2022/13322-5.
**ACKNOWLEDGMENTS**
Eleanor Burke (UK Met Office) for original JULES-ES simulations.
Tristan Quaife (University of Reading) for supervisor support.
Eddy Roberton (UK Met Office) for support and discussion on this research.

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
