# Peer review of "FLAME 1.0: a novel approach for modelling burned area in the Brazilian biomes using the Maximum Entropy concept"

_EGUsphere, 2024_

## Referee Comment (RC2)

Review: egusphere-2024-1775

In this manuscript, the authors describe a clever adjustment of a species distribution model to make it work for burned area. They apply it to the different biomes in Brazil, generating parameter set distributions that can be used to quantify uncertainty in burned area drivers, for both natural and non-natural fire as well as their combination. The authors then go through the biomes discussing the various ways they are similar or different in terms of burned area drivers.

I think this is quite a cool method and application, but the paper itself could use some work in terms of organization, presentation, and clarification. I also have some substantive critiques of the work itself—nothing disqualifying, but still things that should at least be discussed. For that reason I'm recommending major revisions and would be happy to review the next version.

Substantive critiques
- Lumping forest plantation in with croplands seems like an odd choice, considering that drivers of fire in those might be very different. Why did you do this?
- Why didn't you consider deforestation? The use of fire to clear forest is an important driver of burned area.
- Consider separating the southeast Amazon (where there is the most land use and deforestation) from the rest to avoid overestimation of NON burned area. If not, explain why, and discuss it in more depth than the one sentence at lines 855-858.
- Optimizing over 2002-2009 might result in parameters that don't work well for 2010-2019, since those two periods have very different deforestation dynamics. This should be discussed, perhaps in the part of the Discussion where you mention how random partitioning of the data into training and testing datasets can cause problems. The issues you raise there are valid, but it would have avoided this problem!

Suggestions re: presentation
- The last two paragraphs of the Introduction focus too much on fire as a negative force; some Brazilian ecosystems rely on fire for their continued existence! What would happen to the Cerrado if it never burned? Indeed, the authors acknowledge this early in the Methods.
- Fig. 1:
  - Consider regridding the natural/non subplot to 0.5°, showing fraction of each per gridcell. This would make it easier to compare to the maps in the B part.
  - (Also other figures) The bit in the center of this screenshot is mostly just a black mass, obscuring the data you're trying to show there. Consider decreasing the thickness and/or opacity of biome boundary lines.

[Figure]

- All multi-plot figures should have labels for *each* subplot (a, b, etc. according to the GMD guidelines). (As an example of the problem, see what I had to do in my Fig. 1 comment above.)
- Eq. 1 seems unnecessary.
- Eqs. 1 and 2 should have "i" subscripts on the left-hand side.
- Consider marking in Table 1 the variables that got selected.
- Line 354: mention again here that the optimization was over 2002-2009.
- Fig. 5 is hard to parse. I suggest supplementing it with histograms of simulated burned area for each biome, including a vertical line showing the observed value.
- Table 2: Consider converting this to a figure with boxplots rather than a table.
- Rather than the light-pink to dark-pink color scale used on maps (which are often very hard to distinguish), consider something that has different colors. E.g., viridis: https://matplotlib.org/stable/users/explain/colors/colormaps.html
- Fig. 6: As with Fig. 6, consider supplementing with histograms.
- 468-494: This would really benefit from a figure, with for each region either a bar graph showing mean bias or a box plot showing the distribution of biases. It's easier to get information from a single figure than from three paragraphs of text.
- Figs. 8-10
  - $10^{th}/90^{th}$ percentile potential color bars should be symmetrical around zero
  - Likelihood potential color bar: 0-20 and 20-40 are hard to distinguish
- Sect. 3.2: Again, the very verbose explanations here would be very much helped by figures like a bar plot showing, for each region and ALL/NAT/NON, the fraction where it: sees more burning with real values than the median, less burning, etc. This is 8 pages of pure text that is at best hard to get any coherent patterns from, and at worst (as it is for me) actually impossible to focus on well enough to even read. With figures, you could then limit text to only the results that are somehow interesting. (The maps are not in and of themselves good summaries of the regional patterns, because it's hard to judge total area in each category.)
- 803-819: This seems to fit more in an Introduction or maybe Methods section, as it's not really tied in with the results at all. How do your results inform what you've written here?

Clarification needed
- Please replace the use of "MaxEnt" with "maximum entropy" when talking about the concept. This would avoid ambiguity given the species distribution model called MaxEnt that the authors discuss.
- 36-38: Unclear
- 106-107: This needs explanation. How were "negative impacts" defined? And does this number properly account for land that was burned multiple times? The link provided in the citation does not answer these questions.
- "Fires reaching" terminology is confusing. E.g., lines 132-133: "fires reaching natural vegetation (NAT) and fires reaching non-natural vegetation (NON)"—it sounds like you're looking at individual fires, but what about fires that burned both? In reality I don't think you're talking about individual fires, because you probably wouldn't have used the raw MODIS data in that case. I would rephrase lines 132-133 as "burned natural vegetation (NAT) and burned non-natural vegetation (NON)" (and rephrase similar text to match).
- 139-148:

- o It's unclear at this point whether biome is a variable in your model or just something you'll consider when interpreting the results. If the latter, move this to Sect. 2.5. (After looking at Table 1, it looks like it is indeed not actually in the model.)
  - o This bit also doesn't fit with the beginning of the paragraph (land use).
- 197-212:
  - o Justify these metrics *before* describing them, not after.
  - o Are "classes" here just NAT/NON or forest/grassland/crop/etc.?
- Table 1 caption: "Initial" list? I guess this means before removal as described in Sect. 2.2; mention that.
- 225:
  - o Clarify that you removed just one of each pair of highly-correlated variables.
  - o How did you choose which of each pair to remove?
- 238: What do you mean, "Initially"?
- 258-270: Explain that you tested the *combination* of linear and power relationships, and that you did not constrain your parameters *a priori* to require positive or negative relationships.
- 331-333: How is Q parameterized?
- 348-351: How were these definitions of "too wide" and "too narrow" determined?
- 356: Same question about 50%.
- 418-419: Define "uncertainties." Is this just "difference between 10$^{th}$ and 90$^{th}$ percentiles"? And is it 10% (i.e., uncertainty *relative* to the mean/median) or 10 *percentage points*?
- Figs. 5-6, 8-10: Are percentiles here defined based on likelihood? Or is it burned area?
- 454-457: Didn't you already stratify fire based on vegetation type—NAT and NON? Or do you mean *within* those categories?
- For sensitivity tests: Did you always change members of a given group in the same direction? E.g., for Group 1, did you compare Temp–0.05/Precip–0.05, Temp+0/Precip+0, Temp+0.05/Precip+0.05? In that particular case, the perturbations would work against each other, reducing the apparent sensitivity. What you should do is perturb everything in each group so they work *together* in each direction. You may have, but I don't think you actually say that anywhere.
- Figs. 8-10
  - o Sensitivity plots: Is this the relative difference between the +0.05 and –0.05 runs? Why is it always positive?
  - o Where do the likelihood numbers come from? Medium likelihood values (40-60%) being considered "not confidently predictable" is very confusing. This is not the same way that likelihood is treated in e.g. Fig. 6.
- 852-854: Is this something that's *not* reflected in your results? If so, what are the implications of that?
- 897: Remind the reader what variables are in Group 3. And is that number all positive influence? Is there any additional area of negative influence?
- 959-960: This sentence is unclear, especially the second half.
-

Corrections
- 57, 58: in citations, replace semicolons with "and"
- 132: LULC abbreviation not defined.

- Fig. 3 is very low-resolution, and text is too small.
- 288-289: Code must be associated with a DOI and included in Code and Data Availability section.
- The last power term in Eq. 9 is $(m - s)/s$. If, as you say at line 325, $\lim_{s \to \infty} m/s = BF$, then that power should become $BF - 1$, but Eq. 10 has that backwards $(1 - BF)$.
- 379: "student" should be capitalized.
- Throughout: Author names should be in Title Case, not CAPITALS.
- Fig. 5 is very low-resolution.
- Table 2 (if kept as a table and not converted to a figure; see "Suggestions re: presentation") needs to be an actual table, not a screenshot. This is critical for legibility and accessibility.
- Fig. 6
  - Very low-resolution.
  - These are not best and worst likelihoods (i.e., maximum and minimum) as the caption says, but rather 90th and 10th percentile.
- 502: Should "Specifically" be "For example"?
- 931: "Perilous" might not be the right word; I'm not sure what it's trying to say in this context.

---

## Author Comment (AC1)

**Response to Reviewer Comments**

**Reviewer 1:**

The authors present a novel approach to model burned area, combining two frameworks that have previously been used to model species distribution (Maximum Entropy) and fire (Bayesian Inference). Applying FLAME 1.0 to Brazilian biomes, they evaluate the model performance, differentiating fire categories, and analyze the contribution of various predictor variables to burned area. Additionally, they provide a detailed analysis of the uncertainties in the relationship between fire drivers and fire occurrences. This paper presents a very relevant and comprehensive study, and makes a strong case for future research to give greater consideration to regional nuances in fire behavior.

Thank you for taking time to review our work. We greatly appreciate your positive and constructive feedback.

General comments

This work is nicely done, and my comments are mostly editorial. Throughout the manuscript, I would suggest to more carefully differentiate between statistical/ data-driven fire modelling vs process-based modelling (I highlighted some of those occurrences in the specific comments below). Similarly, I would suggest to consistently refer to 'independent variables'/ 'explanatory variables' or 'predictors' as opposed to just calling them 'variables'. These are really minor suggestions, and I provide some more specific comments of similar nature below.

Response: Thank you for your suggestion. We have revised the text to consistently use the term "explanatory variables" throughout the manuscript, ensuring clarity and alignment with standard terminology.

Specific comments

L39: I would suggest to write 'particularly for the Pampas and Pantanal regions'

Response: Thank you for the suggestion. We have revised the text to include 'particularly for the Pampas and Pantanal regions' as recommended.

L58: There is probably no right or wrong with this, but I would suggest to use Terrestrial Biosphere Model (TBM) as an umbrella term for both LSMs and DGVMs

Response: As suggested, the term LSM was replaced by TBM.

L76-81: Referring to Bayesian Inference in relation to fire the should also have a reference - e.g. Kelley et al., 2019?

Response: As suggested, a reference to Kelley et al., 2019 has been added to support the mention of Bayesian Inference in relation to fire.

L98: Maybe you could also link to Bayesian Inference here to highlight that Max Entropy alone is not sufficient to model fire, but the combination of both can. The description of Bayesian Inference is a bit sandwiched between descriptions of Max Entropy as it is anyway - maybe it would work to move that part here? But I don't insist on this suggestion.

Response: Thank you for your suggestion. To first present the explanation of the MaxEnt model followed by the Bayesian inference approach and enhance clarity, we have moved the section. However, it is important to note that Maximum Entropy has demonstrated its capability to independently model fires, as highlighted in the manuscript. In this context, the goal of our approach was to enhance its application and enable the simulation of burned area fractions.

L130:  I think there needs to be a space between 500 and m. How did you regrid the data?

Response: Corrected. The data was regridded by dividing the total burned area within each coarse cell by its total area. An Explanation has been added in the revised manuscript.

L132: You're not consistent when citing Map Biomas throughout the manuscript

Response: We acknowledge your comment and have revised the manuscript to ensure consistent use of the term 'MapBiomas' throughout.

L132: You haven't defined the abbreviation LULC yet

Response: We acknowledge your comment and have revised the manuscript to refer to Land Use and Land Cover as LULC.

L143-148: I would first describe how biomes were defined (i.e. move this part to L139 so that it reads '[...] across different vegetation types. We based our vegetation categorization on Hardesty et al., 2005 [...]' or similar) and then at the end describe the specific biomes you came up with.

Response: Thank you for the suggestion. We have restructured the paragraph as recommended and added a dedicated section (2.1) to provide an explanation of the Brazilian biomes. This adjustment addresses the confusion raised by Reviewer 2 regarding the definition and context of biomes.

L158: Are you also including lagged variables? I wonder whether, if you only use August, September, and October values, you lose memory effects in the fuel properties

Response: We did not explicitly include lagged climate variables in our model runs. Still, the variable representing carbon in dead vegetation reflects the accumulation of fuel load over time, thereby incorporating memory effects. However, this study does not account for memory effects related to fuel moisture, which could be considered in future research. And consecutive dry days and soil moisture are provided in our data (Table 1) should anyone wish to use them.

L168: Can you specify whether you use a sub-daily/ daily or monthly timestep for the climate variables?

Response: We used monthly data and have included this information in the manuscript.

L169: 'We obtained soil and vegetation carbon and soil moisture' (?) Later in the manuscript you mention you used dead vegetation carbon. If that is the case, you should mention it here.

Response: Thank you for your comment. We have replaced the term "soil carbon" with "carbon in dead vegetation" to avoid ambiguity. However, we acknowledge that the change was not consistently applied throughout the manuscript. This issue has now been corrected.

L195: How did you interpolate the data? Presumably just a linear interpolation?

Response: The data was linearly interpolated as per Kelley et al. (2019). We have added this information in the revised manuscript.

L197: Maybe rephrase a bit? 'were calculated for each grid cell' or similar?

Response: We acknowledge your comment and have rephrased the manuscript to "Total road density (in m/km²) was calculated for each 0.5-degree grid cell using linear interpolation in the Iris Python package (MET OFFICE, 2023), based on road density data from the Global Roads Inventory Project (GRIP) (MEIJER et al., 2018)".

L197-212: Are the forest metrics constant or do they change in time?

Response: The forest metrics change in time. The data is based on the annual Mapbiomas data explained in the manuscript which were linearly interpolated from annual to monthly timestep.

Table1: How did you calculate the dry days? I'm mostly wondering in terms of the temporal dimension (e.g. do you restart consecutive dry days every August, or [...]?)

Response: The variable "consecutive dry days" represents the number of dry days since the last recorded rainfall, starting from the 1900s. The count is continuous and does not reset. We then calculate the monthly maximum of this running value. We have added this description to the revised manuscript.

L222: I think you could be a bit more precise here: You're assessing the relationships among the predictor variables, not between the predictor variables and fire in this part of your analysis. Of course there are likely non-linear relationships here as well but I'm not sure arguing with the non-linear relationship between predictors AND fire is the best justification here.

Response: Thank you for your comment. The words "and fire" were removed to precisely express that the analysis is only testing the relationship among the variables.

L228-232: Then why did you include lightning in the first place:)

Response: Thank you, that's a fair point. We have deleted lightning from the correlation analysis and included an updated Figure 2. The text referring to this matter has been removed.

L234: I would refer to Table 1 after 'explanatory variables'

Response: We have added a reference to Table 1 in the caption of Figure 2 to clarify the explanatory variables, as follows: "Figure 2: Spearman correlation of the explanatory variables (also see Table 1)...".

L238-240: Did you change the number of predictor variables later on? 'Initially' implies that you did. If you did not, do you expect that changing the number of predictor variables would have a strong impact on your results? If it's not a lot of work it could be interesting to see in the supplement. However, your manuscript is already very comprehensive and if this would be a lot of work that wouldn't add a lot, feel free to ignore my comment.

Response: Thank you for your comment. We tested the model with different numbers of predictors and observed that while uncertainties regarding burned area simulations decreased, uncertainties in the responses of individual variables increased. It would be interesting in future work to explore other functions and incorporate stronger priors when adding more variables which we did not test here. For this study, our primary objective was to document the model, so we kept the list of predictors concise to facilitate a clearer discussion of the results. However, we agree that the optimal number of predictors depends on the specific research questions, and this is an area that warrants further exploration in future studies.

L243: I like that you explain how the predictors relate to fire. Maybe you could include that Tmax and precip relate to fire weather?

Response: Thank you for your suggestion. It has been incorporated in the revised manuscript.

L245: See my comment earlier: Did you exclusively use carbon in dead vegetation? If so, how did you derive it (or is it a direct JULES output?)

Response: The variable carbon in dead vegetation combines the decomposable plant material and resistant plant material carbon pools, which are direct outputs from JULES (Clark et al. 2011; Burton et al. 2019).

L248: Can you specify how you aggregated the explanatory variables over time? Does group 2 vary with time or is it constant?

Response: In the figure, the values represent the mean for August, September, and October from 2002 to 2019. However, in the model, we used all data points within these months for sampling—2002-2009 for training and 2010-2019 for evaluation. The edge density variable was calculated annually and then linearly interpolated to a monthly time step. In contrast, road density lacks a time dimension, remaining constant over time. We added a column to Table 1 to indicate temporal availability and included this information.

L463-465: I think it would be useful to give more detail on what the different columns (ALL, NAT, NON) depict in the figure caption so it is easier to understand the figure on its own.

Response: Thank you for the suggestion. We have included more detailed explanations for the columns (ALL, NAT, NON) and rows in the figure caption to enhance clarity and ensure the figure is self-explanatory as suggested. This adjustment has also been applied to Figure 5 and Figure 7(Now Figure 8) captions.

L468-494: You mention a lot of values for the mean bias in these two paragraphs, and I'm not sure where they are coming from? Are they listed anywhere?

Response: They originated from the bias analysis, although we did not explicitly list them in the manuscript. We have now included a figure(7) summarizing these results as also suggested by reviewer 2.

L521-524: I would suggest to describe the interpretation of all colors in your figures in the figure caption, including why pixels are white - even if it sounds painfully obvious.

Response: We have added the interpretation of all colors in the figures caption.

L536-538: I find it a bit confusing here, and also later on, that you don't specify the direction of the deviation. Is that because your framework doesn't allow that? Intuitively I would expect that values below the median would have a different impact than values higher than the median, but I might have just misunderstood the metric.

Response: The direction of the deviation depends on the original values of the variables. If the original value of a variable was above the median, the analysis examines the effects of the variable on the burned area when it assumes values above the median. In the case of Group 1 (Maximum Temperature and Precipitation), we anticipate that a combination of below-median precipitation and above-median temperature is likely to result in an increase in burned area. However, this same combination may have no impact on burned area in certain locations due to the influence of other factors or because the deviation is not extreme enough. While the framework does not explicitly indicate the direction of the deviation, it helps identify locations where these relationships are more likely to either increase or decrease burned area. We hope this explanation helps to clarify the matter and we have added that the deviation could go both directions in the revised manuscript.

L561-562: 'The remaining 4.53% of the area remains uncertain' this is a bit vague. Can rephrase it?

Response: We have rephrased the sentence to ensure clarity.

L683-684: I found this also a bit unclear

Response: Thank you for pointing this out. We have clarified the text to better describe the responses to Group 3 in the ALL category. Additionally, we added a more detailed explanation about the interpretation of the likelihood maps in the methods section 2.5 (now 2.6).

Figures 8 - 10: Could you mention either in the figure itself or in the figure title (at least in Figure 8) what the different groups are again?

Response: We acknowledge your comment and have revised the Figure caption to include the variables in each group.

L777: Here for example, I would refer to 'process-based' fire models here rather than 'conventional fire models' given you're citing two papers looking at process-based fire models coupled (also in L283 - 'global fire models' can be process-based or statistical or data-driven [...])

Response: We have changed the citations as the idea here is to use a general term and not only process-based models. For example, INFERNO and MC2 are classified as "empirical models" distinct from "process-based models" in Hantson et al. (2016), and do not represent the fire stochasticity. In L823 (We are assuming) we have replaced global fire models for process-based fire models.

L821-833: This paragraph is a bit unorganized and I'm not clear which aspect you're focusing on here: That your model is able to better capture temporal variability in fire patterns, or that it has a better regional representation? I assume the last one, but L822 sounds like you tested how well FLAME did in capturing temporal variability. Maybe I missed but did you really test the performance over time? You then also make a bit of a jump to coupled Earth System Modelling in my opinion but the point you make is of course very valid. I would suggest to rewrite it to something like 'Additionally, while fire-enabled Earth System Models can integrate feedback mechanisms between land and atmosphere, therefore enabling the evaluation of inter-variable effects, offline global fire models do not. Similarly, FLAME is not designed to [...]' or something along those lines.

Response: Thank you for your comment and suggestion. You are correct that we did not evaluate FLAME's performance over time. To address this, we have replaced the term "high burning events" with "high burned area" to eliminate ambiguity regarding temporal variability. Additionally, we have incorporated your suggested lines into the text, as they enhance the clarity and quality of the paragraph.

L867-868: 'Despite being a combination with land use' - this is sounds a bit confusing

Response: Thank you for the suggestion. The sentence has been revised to clarify the role of land use changes alongside the extreme climatic conditions observed in 2020.

L880: Would suggest to rewrite to '[...] reaching a level of fragmentation that impedes forest fires from spreading'

Response: Thank you for your suggestion. We have revised the sentence to reflect the idea that forest edges may act as firebreaks, potentially limiting the spread of fires due to the reduction of aboveground biomass near these edges.

L897: I think it would be nice to repeat here what Group 3 is, the way you did in L847 and L871 with Group 1 and 2

Response: Thank you for the suggestion. We have added a description of Group 3, consistent with how we addressed Groups 1 and 2.

L918: 'alternative metrics' - can you give an example?

Response: The approach we used to assess the response of explanatory variables is novel, but alternative methods could also be considered. FLAME was developed as part of a PhD thesis, with one chapter applying the model to the Pantanal biome using targeted variables and additional analyses. In this application, we visualized potential and sensitivity responses as response surfaces, aiming to identify combined thresholds of specific variables that lead to increased burned area. Future research could focus on developing more precise methods for calculating these thresholds. Additionally, a metric specifically targeting extreme fire events could be developed and tested within the model framework.

For reference, the thesis can be accessed in the following link:

http://mtc-m21d.sid.inpe.br/col/sid.inpe.br/mtc-m21d/2024/04.04.17.26/doc/thisInformationItemHomePage.html

L924-926: Why was land use change not included?

Response: We chose to exclude deforestation data because it is not a major driver of burned area across all Brazilian biomes, and we aimed for a general approach. For instance, fires in Pantanal are more strongly associated with the flood pulse levels (Damasceno-Junior et al., 2023) combined with human activities but not necessarily deforestation. In addition, in the Atlantic Forest, deforestation is not strongly linked to present day fire activity (De Praga Baião et al., 2023). Even in the Amazon, fires are not always directly associated with increased deforestation (De Oliveira et al., 2023). However, incorporating deforestation as a variable in future studies, particularly in the Amazon and Cerrado, could provide interesting insights.

L943: Would suggest to rewrite to 'Understanding the factors that drive fires [...]' or similar

Response: We have revised the text as suggested to ensure clarity.

**Reviewer 2**

In this manuscript, the authors describe a clever adjustment of a species distribution model to make it work for burned area. They apply it to the different biomes in Brazil, generating parameter set distributions that can be used to quantify uncertainty in burned area drivers, for both natural and non-natural fire as well as their combination. The authors then go through the biomes discussing the various ways they are similar or different in terms of burned area drivers. I think this is quite a cool method and application, but the paper itself could use some work in terms of organization, presentation, and clarification. I also have some substantive critiques of the work itself—nothing disqualifying, but still things that should at least be discussed. For that reason I'm recommending major revisions and would be happy to review the next version.

Thank you for reviewing our work. We deeply appreciate your insightful comments, which have significantly improved our manuscript.

**Substantive critiques**

·   Lumping forest plantation in with croplands seems like an odd choice, considering that drivers of fire in those might be very different. Why did you do this?

Response: Thank you for your comment. We considered that forest plantations and mechanized agriculture share a key similarity in that both typically avoid the use of fire in their management practices. In this sense, they can be considered analogous, particularly as the cropland class in our model does not distinguish between small-scale and large-scale mechanized farming systems. In addition, compared to croplands, forest plantations represent only a small fraction of most biomes. However, we acknowledge that we could consider excluding them from future analyses to avoid any interference in the results. There are, in fact, many types of fire that could be separated depending on the analysis one may want to conduct (see response to deforestation below). As per the aims and scope of GMD, this paper intends to document the modelling framework that would allow other studies to do this rather than perform analyses on all fire types or all vegetation type breakdown. This adjustment will be kept in mind for future research to refine our approach further.

·   Why didn't you consider deforestation? The use of fire to clear forest is an important driver of burned area.

Response: Deforestation is indeed a significant driver of fire in many regions, particularly in the Amazon . However, it is crucial to recognize that it is not the only human-caused fire issue in Brazil, nor is it the primary driver in other biomes (as also addressed in our response to Reviewer 1). In regions like the Pantanal, fires are frequently linked to wetland drainage, which poses a considerable threat to biodiversity and the livelihoods of local communities. In the Atlantic Forest, deforestation is not closely associated with current fire activity. While deforestation is an important factor, it represents only one of many issues associated with fires in Brazil. One potential application of our model is to explore the relationship between

deforestation and fires in future work. The primary aim of this study, however, was to document the model, rather than to analyze all the drivers of fires across the country.

·       Consider separating the southeast Amazon (where there is the most land use and deforestation) from the rest to avoid overestimation of NON burned area. If not, explain why, and discuss it in more depth than the one sentence at lines 855-858.

Response: We understand your comment, but ideally, this separation should be addressed in future studies focused on the Amazon, as the current paper is already quite extensive, and its primary objective is to document the model. While this limitation may introduce some degree of overestimation of non-burned areas in the southeast Amazon, we believe that the model's overall performance provides valuable insights into fire dynamics. Additionally, the Cerrado also includes distinct regions of natural vegetation and deforested areas, where this approach could similarly be applied to improve regional estimates in future research. Expanding on this in future work would enhance the model's applicability to diverse ecosystems while maintaining clarity in the current study's scope. We have expanded this idea in the revised manuscript.

·       Optimizing over 2002-2009 might result in parameters that don't work well for 2010-2019, since those two periods have very different deforestation dynamics. This should be discussed, perhaps in the part of the Discussion where you mention how random partitioning of the data into training and testing datasets can cause problems. The issues you raise there are valid, but it would have avoided this problem!

Response: Thank you for your comment. This approach was intentionally designed to test the model's ability to capture alternative dynamics and simulate scenarios that may not be present in the observed data. When the same dataset is partitioned into training and validation periods, it becomes challenging to determine whether the model is genuinely capturing the phenomenon or merely replicating patterns learned from the training data. Furthermore, if the model is intended for future projections, and development of this model is intended for applications such as the State of Wildfires report where uncertainty estimates on future projections are critical (Jones et al. 2024), it must be capable of performing with unseen data, where the underlying dynamics could differ significantly. We have added this idea in the revised manuscript.

**Suggestions re: presentation**

·       The last two paragraphs of the Introduction focus too much on fire as a negative force; some Brazilian ecosystems rely on fire for their continued existence! What would happen to the Cerrado if it never burned? Indeed, the authors acknowledge this early in the Methods.

Response: Thank you for your observation. We have revised the paragraphs to better acknowledge the dual role of fire in Brazilian ecosystems, highlighting its ecological importance in fire-adapted systems such as grasslands and savannas while also emphasizing the risks posed by increasing fire frequency. This discussion serves to underscore the delicate balance required to maintain ecosystem structure and composition, as even fire-adapted ecosystems can suffer significant negative impacts when fire regimes are altered beyond their natural thresholds.

Fig. 1:

o  Consider regridding the natural/non subplot to 0.5°, showing fraction of each per gridcell. This would make it easier to compare to the maps in the B part.

Response: We acknowledge that would be easier to compare and we have updated the figure accordingly.

o  (Also other figures) The bit in the center of this screenshot is mostly just a black mass, obscuring the data you're trying to show there. Consider decreasing the thickness and/or opacity of biome boundary lines.

Response: We have decreased the thickness to 0.1; however, it is only noticeable when zoomed in. The complexity and diversity of these two biomes, particularly in transitional areas (ecotones) where the biomes overlap, make establishing clear boundaries more difficult, leading to greater irregularity.

· All multi-plot figures should have labels for *each* subplot (a, b, etc. according to the GMD guidelines). (As an example of the problem, see what I had to do in my Fig. 1 comment above.)

Response: We have updated all multi-plot figures with the appropriate labels.

· Eq. 1 seems unnecessary.

Response: The equation 1 was removed as suggested.

· Eqs. 1 and 2 should have "i" subscripts on the left-hand side.

Response: Eq.1 was removed. Since the metrics were calculated exclusively for forests, we removed the "i" subscript from both sides of the equation 2 (now eq.1) to avoid any misunderstanding that other classes (e.g., grassland, cropland, etc.) were included. Please also refer to our response regarding the use of the term "class."

· Consider marking in Table 1 the variables that got selected.

Response: An additional column was included to indicate which variables were selected for the analyses.

· Line 354: mention again here that the optimization was over 2002-2009.

Response: We have revised the text to include the period from 2002 to 2009 as suggested.

· Fig. 5 is hard to parse. I suggest supplementing it with histograms of simulated burned area for each biome, including a vertical line showing the observed value.

Response: As it is a documentation paper we believe this figure should be in the main text. However, we generated the histograms and supplemented as suggested.

· Table 2: Consider converting this to a figure with boxplots rather than a table.

Response: The table has been converted to a figure(6) as suggested.

· Rather than the light-pink to dark-pink color scale used on maps (which are often very hard to distinguish), consider something that has different colors. E.g., viridis: https://matplotlib.org/stable/users/explain/colors/colormaps.html

Response: We acknowledge that the figure is not easy to distinguish values which is also due to low variation of values among the plot, especially the 90th percentile. We changed the colors and supplemented as suggested.

· Fig. 6: As with Fig. 6, consider supplementing with histograms.

Response: Supplemented as suggested.

· 468-494: This would really benefit from a figure, with for each region either a bar graph showing mean bias or a box plot showing the distribution of biases. It's easier to get information from a single figure than from three paragraphs of text.

Response: As suggested, we included a bar graph showing the mean bias and the 10th-90th percentiles (Figure 7).

Figs. 8-10

o 10th/90th percentile potential color bars should be symmetrical around zero

Response: The figures 8-10 (now figures 10-12) were modified as suggested.

o Likelihood potential color bar: 0-20 and 20-40 are hard to distinguish

Response: The color of the 20-40 was changed to blue to better distinguish between the 0-20 class.

· Sect. 3.2: Again, the very verbose explanations here would be very much helped by figures like a bar plot showing, for each region and ALL/NAT/NON, the fraction where it: sees more burning with real values than the median, less burning, etc. This is 8 pages of pure text that is at best hard to get any coherent patterns from, and at worst (as it is for me) actually impossible to focus on well enough to even read. With figures, you could then limit text to only the results

that are somehow interesting. (The maps are not in and of themselves good summaries of the regional patterns, because it's hard to judge total area in each category.)

Response: You made a fair point. As suggested one figure summarizing these results were added and the text was reduced to improve readability.

· 803-819: This seems to fit more in an Introduction or maybe Methods section, as it's not really tied in with the results at all. How do your results inform what you've written here?

Response: We understand your perspective. This section focuses on the model, and our intention was to draw a comparison with the most widely used maximum entropy model employed for fire analysis. Our goal is to highlight the advantages of our approach. While we have condensed this paragraph, we believe it is important to retain this concept within the discussion.

Clarification needed

· Please replace the use of "MaxEnt" with "maximum entropy" when talking about the concept. This would avoid ambiguity given the species distribution model called MaxEnt that the authors discuss.

Response: Thank you for pointing this out. We have revised the manuscript to replace "MaxEnt" with "maximum entropy" as suggested to avoid any ambiguity.

· 36-38: Unclear

Response: Thank you for your comment. This statement is wrong and was removed from the abstract.

· 106-107: This needs explanation. How were "negative impacts" defined? And does this number properly account for land that was burned multiple times? The link provided in the citation does not answer these questions.

Response: Thank you for your point. The term "negative impacts" was not used properly in the sentence as the dataset does not make this distinction. To correct that, the word "negatively" and the sentence "reflecting a need for effective and adaptive fire management strategies" were removed. The number refers to land that burned at least once between 1985-2022. This information was added to improve clarity.

· "Fires reaching" terminology is confusing. E.g., lines 132-133: "fires reaching natural vegetation (NAT) and fires reaching non-natural vegetation (NON)"—it sounds like you're looking at individual fires, but what about fires that burned both? In reality I don't think you're talking about individual fires, because you probably wouldn't have used the raw

MODIS data in that case. I would rephrase lines 132-133 as "burned natural vegetation (NAT) and burned non-natural vegetation (NON)" (and rephrase similar text to match).

Response: Thank you for pointing that out. We have revised the text to "burned areas in natural vegetation (NAT)" and "burned areas in non-natural vegetation (NON)" instead. This change has been applied consistently throughout the manuscript to ensure clarity and coherence.

139-148:

o It's unclear at this point whether biome is a variable in your model or just something you'll consider when interpreting the results. If the latter, move this to Sect. 2.5. (After looking at Table 1, it looks like it is indeed not actually in the model.)

Response: A biome is a distinct geographical region characterized by specific climate conditions, vegetation, and wildlife. Brazil is home to six biomes classified by the Brazilian Institute of Geography and Statistics (IBGE - https://www.ibge.gov.br/apps/biomas/#/home). We optimized and evaluated a separate model to each biome (also see section 2.4[now 2.5]). To ensure there is no misunderstanding, we have included a dedicated section (2.1) to explain our unit of analysis.

o This bit also doesn't fit with the beginning of the paragraph (land use).

Response: We have moved the section as explained in the previous comment.

197-212:

o Justify these metrics *before* describing them, not after.

Response: The sentences were reorganized as suggested.

o Are "classes" here just NAT/NON or forest/grassland/crop/etc.?

Response: Refers to forest/grassland/etc. For NAT/NON we are using the term "category". However, in this case, we only calculated these metrics for the forest class so to avoid ambiguity I have removed the term "class" and explicitly added "forest edge density" and "number of forest patches".

Table 1 caption: "Initial" list? I guess this means before removal as described in Sect. 2.2; mention that.

Response: We acknowledge your comment and have clarified the caption to indicate that the list refers to the explanatory variables prior to the removal process described in Section 2.2 (now 2.3).

225:

o Clarify that you removed just one of each pair of highly-correlated variables.

Response: Thank you. We have revised the text to clarify the removal of some of the explanatory variables.

O How did you choose which of each pair to remove?

Response: To clarify, we removed one variable from each pair of highly correlated variables based on the Spearman's correlation matrix, using a threshold of 0.6 (or higher). The decision on which variable to exclude was guided by prior knowledge of the variables' relationships with burned area, as well as their ecological and practical relevance to our study. Specifically, variables that were deemed less directly related to the research objectives or had weaker theoretical support were excluded. To improve clarity, we have revised the text as follows: "We identified explanatory variables with strong correlations using the Spearman's correlation matrix and removed one variable from each highly correlated pair (threshold higher than 0.6). The choice of which variable to remove was informed by prior knowledge of their relationships with burned areas and their relevance to our study".

238: What do you mean, "Initially"?

Response: We appreciate the reviewer's comment and acknowledge that the term "Initially" may have caused confusion. The intention was to clarify that, at this stage of the analysis, 7 explanatory variables were selected from the original 18 based on their correlation and relevance. So, we have revised the sentence for clarity and replaced it with "At this stage" to more accurately convey the intended progression of the analysis.

258-270: Explain that you tested the *combination* of linear and power relationships, and that you did not constrain your parameters *a priori* to require positive or negative relationships.

Response: The explanation was added in the revised manuscript..

331-333: How is Q parameterized?

Response: The parameter Q was set as a log-normal distribution with a $\mu$ of 2 and a $\sigma$ of 1. This explanation was added in the revised manuscript.

348-351: How were these definitions of "too wide" and "too narrow" determined?

356: Same question about 50%.

Response: We are using the 10th-90th percentile range, which inherently excludes 20% of the data (the portion needed to complete 100%). If more than 20% of the observations fall outside this range, it indicates that the model has failed to accurately capture the intended 10th-90th percentile range, making it too narrow. Conversely, if the observations are overly concentrated around the middle of the distribution, it suggests that the model overestimates the spread of the data, resulting in an overly wide uncertainty range. We hope this clarifies the question and we have added this description in the text.

418-419: Define "uncertainties." Is this just "difference between 10th and 90th percentiles"? And is it 10% (i.e., uncertainty *relative* to the mean/median) or 10 *percentage points*?

Response: In this context, the uncertainties refer to the difference between the 10th and 90th percentiles of the simulated burned area distribution. This range captures the central 80% of the simulated values, excluding the lowest 10% and highest 10% of the data, which represent extreme values. We pick 10th to 90th as, typically, the few Bayesian fire modelling studies (Kelley et al. 2019, 2021; Jones at al. 2024) tend to make one-tail comparisons. However, depending on the application, other uncertainty metrics might be appropriate (5-95th, IQR etc). The idea is to provide an estimate of the variability within the model outputs. We have added this description to section 2.5 (now 2.6) in the methods section.

Figs. 5-6, 8-10: Are percentiles here defined based on likelihood? Or is it burned area?

Response: The percentiles in Figures 5 and 8–10 (now 10-12) refer to the burned area. In contrast, the percentiles in Figure 6 (now Figure S.3) are based on the likelihood of the observations given the model, as described in Equations 13 (now eq. 12) and 14 (now eq. 13). We had previously referred to Figure 6 (now Figure S.3) incorrectly in the text but have now corrected this error.

454-457: Didn't you already stratify fire based on vegetation type—NAT and NON? Or do you mean *within* those categories?

Response: Yes, we already stratified fire based on vegetation type (NAT and NON). The purpose of this sentence was to acknowledge that this strategy worked well for the Amazonia. We have revised the sentence to clarify this point.

For sensitivity tests: Did you always change members of a given group in the same direction? E.g., for Group 1, did you compare Temp–0.05/Precip–0.05, Temp+0/Precip+0, Temp+0.05/Precip+0.05? In that particular case, the perturbations would work against each other, reducing the apparent sensitivity. What you should do is perturb everything in each group so they work *together* in each direction. You may have, but I don't think you actually say that anywhere.

Response: The variables in each group were perturbed individually, and the sensitivity of each was combined using the standard definition of a gradient in multi-dimensional space. The perturbations therefore do not work against each other. We have clarified this and added a full derivation in the revised m/s.

Figs. 8-10

o  Sensitivity plots: Is this the relative difference between the +0.05 and –0.05 runs? Why is it always positive?

Response: The sensitivity is defined as the magnitude of the gradient and therefore does not have a defined direction.

o Where do the likelihood numbers come from? Medium likelihood values (40-60%) being considered "not confidently predictable" is very confusing. This is not the same way that likelihood is treated in e.g. Fig. 6.

Response: The likelihood represents the percentage of the modeled distribution that indicates an increase in burning in each biome, or, in other words, how likely it is that the potential response is greater than zero. In the methods section, we previously referred to these as "agreement maps," which we acknowledge that caused confusion. We have now clarified this by referring to them as "likelihood maps." Medium likelihood values (40–60%) indicate that, in those regions, the model predicts an approximately equal likelihood of the variables leading to either increased or decreased burned area. As a result, we cannot draw definitive conclusions in these cases. This explanation was added to section 2.6 (now 2.7).

.
852-854: Is this something that's *not* reflected in your results? If so, what are the implications of that?

Response: In this analysis, we are testing the model's response to marginal climate variations, which suggests that natural vegetation exhibits reduced sensitivity under small changes. However, previous studies have shown that this reduced sensitivity does not hold when the variations are extreme, highlighting the heightened vulnerability of natural vegetation under such conditions. We have clarified this in the text.

897: Remind the reader what variables are in Group 3. And is that number all positive influence? Is there any additional area of negative influence?

Response: We added the variables within group 3 as suggested. While some areas show negative influence, we chose not to discuss every value identified, as this would result in an overly lengthy discussion.

·959-960: This sentence is unclear, especially the second half.

Response: We have rephrased this sentence to: "None of the groups drive huge changes in burned area in the Atlantic Forest. However, since this biome is fire-sensitive, even small changes in burned area can have a substantial impact on its ecosystems."

Corrections

57, 58: in citations, replace semicolons with "and"

Response: Thank you for pointing that out. We have revised the manuscript to ensure consistency throughout.

132: LULC abbreviation not defined.

Response: We have revised the manuscript to define Land Use and Land Cover as LULC.

Fig. 3 is very low-resolution, and text is too small.

Response: The resolution was improved and the text size increased.

288-289: Code must be associated with a DOI and included in Code and Data Availability section.

Response: The code and data are associated with a DOI as referenced in the Code and Data Availability section. The previous reference to the GitHub repository has been removed and replaced with the Zenodo citation.

The last power term in Eq. 9 is $(m - s)/s$. If, as you say at line 325, $\lim m/s = BF$, then $s \to \#$ that power should become $BF - 1$, but Eq. 10 has that backwards $(1 - BF)$.

Response: There was a typo in equation 8 (now eq.7) & 9 (now eq.8). The last power in equation 8 (now eq.7) should be s-m, and equation 9 (now eq.8), (s-m)/s. This has been corrected in the revised manuscript.

379: "student" should be capitalized.

Response: We have corrected "student" to be capitalized as "Student".

Throughout: Author names should be in Title Case, not CAPITALS.

Response: Thank you for pointing that out. We have revised the manuscript to ensure consistency throughout.

Fig. 5 is very low-resolution.

Response: The resolution was improved.

Table 2 (if kept as a table and not converted to a figure; see "Suggestions re: presentation") needs to be an actual table, not a screenshot. This is critical for legibility and accessibility.

Response: The table has been converted into a figure.

Fig. 6

o   Very low-resolution.

Response: The resolution was improved.

o   These are not best and worst likelihoods (i.e., maximum and minimum) as the caption says, but rather 90$^{th}$ and 10$^{th}$ percentile.

Response: We have rephrased to "Lower-" and "Upper-decile performance".

502: Should "Specifically" be "For example"?

Response:   Thank you for pointing that out. We have revised the text and replaced "Specifically" with "For example" for better clarity.

931: "Perilous" might not be the right word; I'm not sure what it's trying to say in this context.

Response: Thank you for noticing this mistake. We meant to say "Previous" and not "Perilous". The word has been corrected.

References

BURTON, C., BETTS, R., CARDOSO, M., FELDPAUSCH, T.R., HARPER, A., JONES, C.D., KELLEY, D.I., ROBERTSON, E. AND WILTSHIRE, A., 2019. Representation of fire, land-use change and vegetation dynamics in the Joint UK Land Environment Simulator vn4. 9 (JULES). Geoscientific Model Development, 12(1), pp.179-193.

CLARK, D.B., MERCADO, L.M., SITCH, S., JONES, C.D., GEDNEY, N., BEST, M.J., PRYOR, M., ROONEY, G.G., ESSERY, R.L.H., BLYTH, E. AND BOUCHER, O., 2011. The Joint UK Land Environment Simulator (JULES), model description–Part 2: carbon fluxes and vegetation dynamics. Geoscientific Model Development, 4(3), pp.701-722.

DAMASCENO-JUNIOR, G.A., PEREIRA, A.D.M.M., OLDELAND, J., PAROLIN, P. AND POTT, A., 2022. Fire, flood and Pantanal vegetation. In Flora and vegetation of the pantanal wetland (pp. 661-688). Cham: Springer International Publishing.

DE OLIVEIRA, G., MATAVELI, G., STARK, S.C., JONES, M.W., CARMENTA, R., BRUNSELL, N.A., SANTOS, C.A., DA SILVA JUNIOR, C.A., CUNHA, H.F., DA CUNHA, A.C. AND DOS SANTOS, C.A., 2023. Increasing wildfires threaten progress on halting deforestation in Brazilian Amazonia. Nature Ecology & Evolution, 7(12), pp.1945-1946.

DE PRAGA BAIÃO, C.F., SANTOS, F.C., FERREIRA, M.P., BIGNOTTO, R.B., DA SILVA, R.F.G. AND MASSI, K.G., 2023. The relationship between forest fire and deforestation in the southeast Atlantic rainforest. Plos one, 18(6), p.e0286754.

KELLEY, D.I.; BISTINAS, I.; WHITLEY, R.; BURTON, C.; MARTHEWS, T.R.; DONG, N. How contemporary bioclimatic and human controls change global fire regimes. Nature Climate Change, v.9, p.690-696, 2019.
KELLEY, D.I.; BURTON, C.; HUNTINGFORD, C.; BROWN, M.A.; WHITLEY, R.; DONG, N. Low meteorological influence found in 2019 Amazonia fires. Biogeosciences Discussions, p.1-17, 2021.

JONES, M.W., KELLEY, D.I., BURTON, C.A., DI GIUSEPPE, F., BARBOSA, M.L.F., BRAMBLEBY, E., HARTLEY, A.J., LOMBARDI, A., MATAVELI, G., MCNORTON, J.R. AND SPULER, F.R., 2024. State of wildfires 2023–2024. Earth System Science Data, 16(8), pp.3601-3685.

---

## Author Response (AR2)

Dear Dr. Rabin,

Thank you for your kind response and for your considerations. Please see below a point-by-point response to your comments.

- Reviewer 1 has a good point about the usefulness of lagged climate variables for fuel moisture. It's fine to note that this could be considered in future work, but please make sure to discuss this.

I have added a sentence about this matter in the discussion (section 4.3).

- You gave a very interesting response to Reviewer 1's comment about changing the number of predictor variables. Please mention this in the Discussion.

Thank you. I have mentioned this in the section 4.3 of the discussion.

- Please add to the Discussion a few sentences of your response to Reviewer 1 re: L918 in the original manuscript (clarifying "alternative metrics").

Added.

- You provided good, thoughtful responses to Reviewer 2 about (a) lumping forest plantations and cropland and (b) not including deforestation. Please add this reasoning to the Methods and/or Discussion.

Thank you. I have added the explanation of lumping forest plantations and cropland in the methods (section 2.2) and the explanation about deforestation in the discussion (section 4.3).

Other
- "Amazonia" should never have "the" in front of it; it's like saying "the South America." It would be okay to say "the region of Amazonia" or "the Amazonian region", just as it would be okay to say "the continent of South America" or "the South American continent".

Thank you for the correction. We have revised the manuscript accordingly.

- Zooming in some of the figures reveals what seems like JPEG compression artifacts. If this is true, please replace with the formats as requested in the GMD Guidelines for Authors (https://www.geoscientific-model-development.net/submission.html#figurestables)—JPEGs should only be used for photos. Maps should be in PNG and bar graphs etc. should be in PDF. (Make sure to not just convert your existing files to the new formats, as that would preserve the artifacts. Instead, generate new files from your analysis scripts.)

The maps are in PNG format; however, upon zooming in on the document, I noticed compression artifacts. I believe this issue occurred during the process of adding the

figures to the document. For the final files, as they will be uploaded separately, this problem will not occur.

- Please refer to Fig. 5 in the captions of Figs. S1 and S2 (and vice versa).

Added.

- Fig. 6: (1) Please mention that the lower bound represents the worst performance, the upper bound represents the best, and the dot represents the mean. (2) "Decline" in the caption should be "decile."

Corrected.

- Sect. 4 title "DISCUSSION" should be "Discussion".

(Line numbers below refer to tracked-changes version.)
- L23: "drivers" should be "drives".

Corrected.

- L39: "the" needed before "Pampas".

Corrected.